# Parameters or Privacy: A Provable Tradeoff Between Overparameterization and Membership Inference

**Jasper Tan**
Rice University

**Blake Mason**
Rice University

**Hamid Javadi**
Rice University

**Richard G. Baraniuk**
Rice University

## Abstract

A surprising phenomenon in modern machine learning is the ability of a highly *overparameterized* model to generalize well (small error on the test data) even when it is trained to memorize the training data (zero error on the training data). This has led to an arms race towards increasingly overparameterized models (c.f., deep learning). In this paper, we study an underexplored hidden cost of overparameterization: the fact that overparameterized models may be more vulnerable to *privacy attacks*, in particular the *membership inference* attack that predicts the (potentially sensitive) examples used to train a model. We significantly extend the relatively few empirical results on this problem by theoretically proving for an overparameterized linear regression model in the Gaussian data setting that membership inference vulnerability increases with the number of parameters. Moreover, a range of empirical studies indicates that more complex, nonlinear models exhibit the same behavior. Finally, we extend our analysis towards ridge-regularized linear regression and show in the Gaussian data setting that increased regularization also increases membership inference vulnerability in the overparameterized regime.

## 1 Introduction

As more machine learning models are being trained on sensitive user data (e.g., customer behavior, medical records, personal media), there is a growing concern that these models may serve as a gateway for malicious adversaries to access the models' private training data [1, 2]. For example, the possibility of performing *membership inference* (MI), the task of identifying whether a specific data point was included in a model's training set or not, can be greatly detrimental to user privacy. In settings like healthcare, knowledge of mere inclusion in a training dataset (e.g., hospital visitation records) already reveals sensitive information. Moreover, MI can enable the extraction of verbatim training data from a released model [1].

The general motivation for this paper is the recent realization that privacy issues around machine learning might be exacerbated by today's trend towards increasingly *overparameterized models* that have more parameters than training data points and so can be trained to memorize (attain zero error on) the training data. Surprisingly, some overparameterized models (e.g., large regression models [3], massive deep networks [4, 5]) generalize extremely well [6, 7]. Limited empirical evidence suggests that overparameterization may lead to greater privacy vulnerabilities [1, 8–11]. However, there has been little to no analytical work on this important problem.

*In this paper, we take the first steps towards an analytical understanding of how the number of parameters in a machine learning model relates to MI vulnerability.* In a theoretical direction, we prove for linear regression on Gaussian data in the overparameterized regime that increasing the number of parameters of the model increases its vulnerability to MI (Theorem 3.2). In a supporting empirical direction, we demonstrate that the same behavior holds for a range of more complex models: a latent space model, a time-series model, and a nonlinear random ReLU features model (Section 5).

We also extend our analysis towards the techniques of ridge regularization and noise addition in Section 4. We first prove that MI vulnerability similarly increases with the number of parameters for ridge-regularized models. Surprisingly, we also demonstrate analytically that additional regularization *increases* this vulnerability in overparameterized linear models. Hence, ridge regularization is not always an effective defense against MI and can even be harmful. Afterwards, we show that the privacy-utility trade-off induced by reducing the number of parameters of a linear regression model is equivalent to that obtained by adding independent Gaussian noise to the output of the model when using all available parameters.

Overall, our analyses show that there are settings where reducing the number of parameters of a highly overparameterized model is a simple strategy to protect a model against MI. As the trend towards increasingly overparameterized machine learning models accelerates, *our results make the case for less overparameterization when privacy is a concern.*

**Related work.** The problem of membership inference has received considerable attention, and we refer the reader to [12] for a survey of the prior art. Many works have demonstrated how deep neural networks can be highly vulnerable to MI attacks (cf., [13–16] for example). Various types of attacks, such as shadow models [13], confidence-based attacks [17, 18], and label-only attacks [19] have risen to further expose privacy vulnerability of machine learning models. Our analysis in this work is based on optimal MI attacks, a framework also employed by other studies of MI [9, 14, 10].

While we study the effect parameterization has on MI vulnerability, several studies examine how other aspects of the machine learning pipeline, such as data augmentation [20], dropout [15], sparsity [21], and ensembles [22] affect MI vulnerability. [16] show that pruning a neural network can defend against MI, lending further experimental support to the link between overparameterization and membership inference. [9] analyzes the relationship between overfitting (as measured by generalization error) and membership inference but does not connect this to the number of parameters. Much recent work has shown though that overparameterization does not always lead to increased generalization error and indeed sometimes can even decrease it [23, 6, 3, 7], suggesting that more work is needed linking overparameterization and membership inference beyond mere generalization error. Despite some empirical evidence that overparameterized models are vulnerable to MI attacks [1, 11], to the best of our knowledge, there has been no theoretical study of the effect of this connection.

Differential privacy (DP) [24] is another popular framework frequently used to study the privacy properties of machine learning algorithms, and models that have DP also enjoy MI guarantees [14]. [25] bounds the minimum dataset size as a function of dimension to ensure privacy, however its results do not extend to data of continuous values (e.g., regression data) as is done in this work and does not account for the randomness in the data generating process, which as argued in [14], is a necessary condition to bound the optimal MI risk.

While our theoretical analysis focuses on linear regression, many works have studied overparameterized linear models as an interpretable and tractable test case of more complex overparameterized models such as deep networks [26, 27, 3, 6, 23], suggesting links of our results to more complex nonlinear models.

**Contributions.** In this paper, we establish the first theoretical connection between overparameterization and vulnerability to membership inference attacks. Our contributions are as follows:

1. We derive the optimal MI accuracy for linear regression on Gaussian data and show that increasing the number of parameters increases the vulnerability to MI.

2. We empirically show for more complex models that increasing the number of parameters also increases MI vulnerability.

3. We theoretically show for overparameterized linear models that increased ridge regularization increases MI vulnerability in the Gaussian data setting.

4. We use our analysis to show that reducing the number of parameters of a linear regression model yields an equivalent privacy-utility trade-off as adding independent noise to the output of a model using all available parameters.

Our code is provided in `https://github.com/tanjasper/parameters_or_privacy`.

## 2  The Membership Inference Problem

We now introduce our notation and formally state the membership inference (MI) problem. Let $D \in \mathbb{Z}^+$ denote the data dimension. Let $z = (\boldsymbol{x}, y) \in \mathbb{R}^D \times \mathbb{R}$ denote a data point, and let $\mathcal{D}$ be a distribution over the data points. Consider a set $\mathcal{S}$ sampled $\mathcal{S} \sim \mathcal{D}^n$ of size $n \in \mathbb{Z}^+$ data points and denote it by $\mathcal{S} = \{z_1, z_2, ..., z_n\} = \{(\boldsymbol{x}_1, y_1), ..., (\boldsymbol{x}_n, y_n)\}$. Let $f_S$ represent a machine learning model obtained by applying a training algorithm $T$ on the dataset $S$. In particular, $f_S$ is a deterministic function in $\mathbb{R}^D \to \mathbb{R}$. Typically, it is a function that minimizes $\sum_{i=1}^n \ell(f(x_i), y_i)$ over a family of functions $f$ for some loss function $\ell : \mathbb{R} \times \mathbb{R} \to \mathbb{R}_{\geq 0}$. We denote $\widehat{y_i} = f_S(x_i)$.

Given a data point $z_0$ and a trained model $f_S$, *membership inference* is the task of identifying if $z_0$ was included in the set $S$ used to train $f_S$. Different MI attacks are characterized by the types of information accorded to the adversary. In this work, we focus on a blind black-box setting where the adversary has access to $\boldsymbol{x}_0$, $f_S(\boldsymbol{x}_0)$, $n$, the training algorithm $T$, and the data distribution $\mathcal{D}$, but not to the ground truth $y_0$, the loss value $\ell(f_S(\boldsymbol{x}_0), y_0)$, or any learned parameters of $f_S$. Thus, the adversary is a function $A : \mathbb{R}^d \times \mathbb{R} \to \{0, 1\}$ such that given $(\boldsymbol{x}_0, f_S(\boldsymbol{x}_0))$, it outputs 0 or 1, representing its prediction as to whether $\boldsymbol{x}_0$ is a member of $S$ based on $f_S(\boldsymbol{x}_0)$.

There are two main interests in studying blind MI wherein the adversary does not have access to the loss value. First, there are many realistic scenarios where the ground truth variable of interest is unknown for the general population, and hence the need for a model $f_S$ to predict it. Second, since we study the interpolation regime where the training loss can be driven down to exactly 0, MI becomes trivial given the loss value: an adversary that predicts any data point that achieves 0 loss as a member will have perfect accuracy. Multiple works in the literature also study this blind setting [18, 28]. We emphasize that our results on the blind adversary lower bound the performance of adversaries that additionally have access to more information.

MI is often defined as an experiment to facilitate precise analysis, and we follow the setup of [9] summarized below.

*Experiment* 1. **Membership inference experiment.** Given a data distribution $\mathcal{D}$, an integer $n$, a machine learning training algorithm $T$, and an adversary $A$, the MI experiment is as follows:

1. Sample $\mathcal{S} \sim \mathcal{D}^n$.

2. Apply the training algorithm $T$ on $\mathcal{S}$ to obtain $f_{\mathcal{S}}$.

3. Sample $m \in \{0, 1\}$ uniformly at random.

4. If $m = 0$, sample a data point $(\boldsymbol{x}_0, y_0) \sim \mathcal{D}$. Else, sample a data point $(\boldsymbol{x}_0, y_0) \in \mathcal{S}$ uniformly at random.

5. Observe the adversary's prediction $A(x_0, f_S(x_0)) \in \{0, 1\}$.

We wish to quantify the optimal performance of the adversary $A$ and turn to the popular *membership advantage* metric defined in [9]. The membership advantage of $A$ is the difference between the true positive rate and false positive rate of its predictions.

**Definition 2.1** ([9]). The *membership advantage* of an adversary $A$ is:
$$\text{Adv}(A) := \mathbb{P}(A(\boldsymbol{x}_0, f_S(\boldsymbol{x}_0)) = 1 \mid m = 1) - \mathbb{P}(A(\boldsymbol{x}_0, f_S(\boldsymbol{x}_0)) = 1 \mid m = 0),$$
where $\mathbb{P}(\cdot)$ is taken jointly over all randomness in Exp. 1.

Note that membership advantage is an average-case metric, as opposed to worst-case metrics considered by other privacy frameworks such as differential privacy. Our analysis is thus focused on average-case privacy leakage, and we do not provide worst-case guarantees.

## 3  Theoretical Results

### 3.1  Optimal Membership Inference Via Hypothesis Testing

In this paper, rather than studying current MI attacks, we study the optimal MI adversary: the adversary that maximizes membership advantage. As such, our analysis and results are not restricted to the current known attack strategies, which are constantly evolving, and instead serve as upper bounds for the performance of any MI attack, now or in the future. The following proposition supplies the theoretically optimal MI adversary.

**Proposition 3.1.** *The adversary that maximizes membership advantage is:*

$$A^*(\boldsymbol{x}_0, f_S(\boldsymbol{x}_0)) = \begin{cases} 1 & \text{if } P(\widehat{y_0} \mid m = 1, \boldsymbol{x}_0) > P(\widehat{y_0} \mid m = 0, \boldsymbol{x}_0), \\ 0 & \text{otherwise}, \end{cases}$$

*where $\widehat{y_0} = f_S(\boldsymbol{x}_0)$ and $P$ denotes the distribution function for $\widehat{y_0}$ over the randomness in the membership inference experiment conditioned on $\boldsymbol{x}_0$.*

As observed by [29, 10], the optimal adversary performs a hypothesis test with respect to the posterior probabilities, with the two hypotheses being:

$$H_0 : \mathcal{S} \sim \mathcal{D}^n, (\boldsymbol{x}_0, y_0) \sim \mathcal{D} \quad \text{and} \quad H_1 : \mathcal{S} \sim \mathcal{D}^n, (\boldsymbol{x}_0, y_0) \sim \mathcal{S},$$

so that maximizing membership advantage is achieved by performing a likelihood ratio test.

## 3.2 Linear Regression with Gaussian Data

We begin our study of overparameterization's effect on MI with linear regression with Gaussian data. We find that in the sufficiently overparameterized regime, increasing the number of parameters increases vulnerability to MI. We denote by $n, p, D$ the number of data points, number of model parameters, and data dimensionality, respectively. Let $n, p, D \in \mathbb{Z}^+$ be given such that $p \leq D$, and let $\sigma > 0$ be given. Consider a $D$-dimensional random vector $\beta \sim \mathcal{N}\left(0, \frac{1}{D}\boldsymbol{I}_D\right)$, representing the true coefficients. We consider data points $(\boldsymbol{x}_i, y_i)$ where $\boldsymbol{x}_i \sim \mathcal{N}(0, \boldsymbol{I}_d)$ and $y_i = \boldsymbol{x}_i^\top \beta + \epsilon_i$, where $\epsilon_i \sim \mathcal{N}(0, \sigma^2)$. Sampling $n$ data points, we denote by $\boldsymbol{X}$ the $n \times D$ matrix whose $i^{\text{th}}$ row is $x_i^\top$ and by $\boldsymbol{y}$ the $n$-dimensional vector of elements $y_i$. Further, let $\boldsymbol{X}_p$ be the $n \times p$ matrix containing the first $p$ columns of $\boldsymbol{X}$. Least squares linear regression finds the minimum-norm $p$-dimensional vector $\hat{\beta}$ that minimizes $\|\boldsymbol{y} - \boldsymbol{X}_p\beta\|_2^2$, which is $\hat{\beta} = \boldsymbol{X}_p^\dagger \boldsymbol{y}$, where $\boldsymbol{X}_p^\dagger$ denotes the Moore-Penrose inverse of $\boldsymbol{X}_p$. Then, for a given feature vector $\boldsymbol{x}_0$, the model predicts $\widehat{y_0} = \boldsymbol{x}_{0,p}^\top \hat{\beta}$, where $\boldsymbol{x}_{0,p}$ is the vector containing the first $p$ elements of $\boldsymbol{x}_0$. In this setting, we can derive the membership advantage of the optimal adversary as a function of the number of parameters $p$ used by the model.

**Theorem 3.2.** *Let $n, p, D \in \mathbb{Z}^+$ be given such that $n + 1 < p \leq D$. Let $\boldsymbol{x}_0$ be a given $D$-dimensional vector. Let $\beta \sim \mathcal{N}(0, \frac{1}{D}\boldsymbol{I}_D), \boldsymbol{x}_i \sim \mathcal{N}(0, \boldsymbol{I}_D), \epsilon_i \sim \mathcal{N}(0, \sigma^2)$, and $y_i = \boldsymbol{x}_i^\top \beta + \epsilon_i$ for $i \in \{1, 2, ..., n\}$. Let $m$ be a variable whose value is either 0 or 1 such that, if $m = 1$, $\boldsymbol{x}_k$ is set to $\boldsymbol{x}_0$ for $k$ chosen uniformly at random from $1, 2, ..., n$. Let $\boldsymbol{X}$ be the $n \times D$ matrix whose rows are $\boldsymbol{x}_1^\top, \boldsymbol{x}_2^\top, ..., \boldsymbol{x}_n^\top$. Finally, let $\widehat{y_0} = \boldsymbol{x}_{0,p}^\top \boldsymbol{X}_p^\dagger \boldsymbol{y}$ where $\boldsymbol{y}$ is the $n$-dimensional vector with elements $y_i$. Then, as $n, p, D \to \infty$ such that $\frac{p}{n} \to \gamma \in (1, \infty)$, we have:*

$$(\widehat{y_0} \mid m = 0, \boldsymbol{x}_0) \sim \mathcal{N}(0, \sigma_0^2) \quad \text{and} \quad (\widehat{y_0} \mid m = 1, \boldsymbol{x}_0) \sim \mathcal{N}(0, \sigma_1^2) \tag{1}$$

*where*

$$\sigma_0^2 := \left(\frac{n}{p}\right)\left(\frac{1}{D} + \frac{1 + \sigma^2 - \frac{p}{D}}{p - n - 1}\right)\|\boldsymbol{x}_{0,p}\|^2 \quad \text{and} \quad \sigma_1^2 := \sigma^2 + \frac{\|\boldsymbol{x}_0\|^2}{D}. \tag{2}$$

*Consider the case when $\sigma_1 > \sigma_0$. The resulting optimal membership inference algorithm $A^*$ is*

$$A^*(\boldsymbol{x}_0, \widehat{y_0}) := \mathbb{1}\left[\widehat{y_0}^2 > \alpha^2\right] \quad \text{where} \quad \alpha := \sqrt{\frac{\sigma_0^2 \sigma_1^2 \log\left(\frac{\sigma_1^2}{\sigma_0^2}\right)}{\sigma_1^2 - \sigma_0^2}}. \tag{3}$$

*with membership inference advantage* Adv*:*

$$\text{Adv}(A^*) = \mathbb{E}_{\boldsymbol{x}_0}\left[2\left\{\Phi\left(\frac{\alpha}{\sigma_0}\right) - \Phi\left(\frac{\alpha}{\sigma_1}\right)\right\}\right], \tag{4}$$

*where $\Phi(\cdot)$ denotes the CDF of the standard normal, and $\alpha, \sigma_0, \sigma_1$ are conditioned on $\boldsymbol{x}_0$.*

*Remark 3.3.* The case where $\sigma_1 < \sigma_0$ occurs when $\gamma$ is small ($p \approx n$). The same membership advantage result holds in this, reversing the roles of $\sigma_0$ and $\sigma_1$ in Adv$(A^*)$ in Eqs. (3) and (4) and reversing the inequality in Eq. (3).

*Remark 3.4.* The above result holds using the asymptotic distributions as $n, p, D \to \infty$. In Lemma C.1 in the Appendix, we derive the non-asymptotic distributions for the predictions of the minimum-norm least squares interpolator, though they cannot be written in closed form.

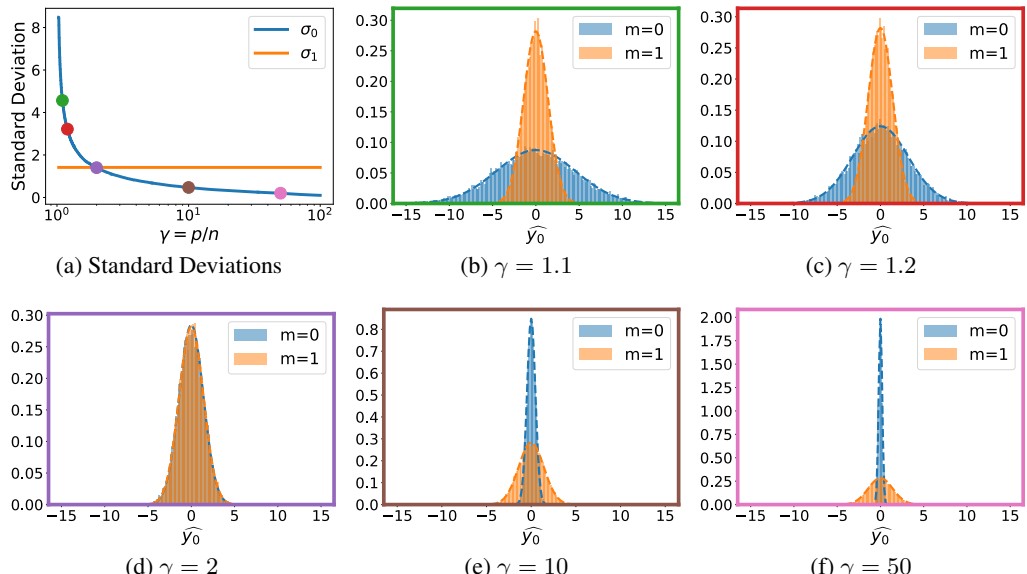

Figure 1: In the highly overparameterized regime, increasing the number of parameters $p$ yields more distinguishable posterior distributions. (a) Standard deviations of model outputs $\widehat{y_0}$ for minimum-norm least squares as a function of parameterization for $\gamma > 1$. (b-f) The Gaussian distributions in Eq. (1) (broken lines), as well as empirical histograms of 20,000 $\widehat{y_0}$ samples for different $\gamma$ values with $n = 400$, $D = 20,000$, $\sigma = 1$. The prediction variance for $m = 1$ stays constant, while the variance for $m = 0$ decreases with increased parameterization, making the distributions easier to distinguish.

In Theorem 3.2, Eq. (1) shows that the posterior distributions of the outputs for test points (when $m = 0$) and training points (when $m = 1$) are both 0-mean Gaussians but with variances $\sigma_0^2$ and $\sigma_1^2$ respectively given in Eq. (2). Recall from Prop. 3.1 that the optimal MI adversary is a likelihood ratio test (LRT) between the distribution of the model's output for new test points (when $m = 0$) and that for reused training points (when $m = 1$). This is reflected in Eq. (3) where we note that $\alpha$ is the standard sufficient statistic for an LRT between two 0-mean Gaussian distributions with deviations $\sigma_0$ and $\sigma_1$. Finally, we compute the membership advantage of this adversary in Eq. (4) by taking an expectation over the random draw of $\boldsymbol{x}_0$, as defined in Experiment 1, noting that $\sigma_0, \sigma_1$, and $\alpha$ are all functions of $\boldsymbol{x}_0$. Membership advantage is defined to be the difference between the true and false positive probabilities in Defn. 2.1. This difference is given by $\Phi(\alpha/\sigma_0) - \Phi(\alpha/\sigma_1)$ for each side of the Gaussian distributions, leading to the expression in Eq. (4).

To understand the implication of the result for MI, first observe that when $m = 1$ and $\boldsymbol{x}_0$ is a training point that is memorized by the model, the variance of the model's output is equal to the variance of the measurement $y_0$ itself independent of $p$. On the contrary, when $m = 0$ and $\boldsymbol{x}_0$ is a test point, the variance of the model's output is *decreasing* with $p$ (Figure 1a). Hence, though the output distribution means stay the same, as the variance for the $m = 0$ case decreases far below that of the $m = 1$ case, an LRT can easily distinguish these two distributions. In the extreme case, suppose $p, D \gg n$, then $\sigma_0^2 \to 0$, while $\sigma_1^2$ remains a nonzero value. Hence $\mathrm{Adv}(A^*) \to 1$.

We confirm our derivations of the output distributions numerically in Figure 1 where we plot empirical histograms of the predicted distributions from Eq. (1) by repeatedly computing the minimum norm least squares solution over 20,000 independent trials[1]. We also plot the standard deviations from Eq. (2) in Fig. 1a. Visually, from the distributions in Figure 1, given $f(\boldsymbol{x}_0)$, MI reduces to identifying whether it is more likely that a sample came from the blue distribution or from the orange with the adversary simply predicting the more likely outcome. These distributions intersect at $\pm\alpha$, so it is sufficient to compare $\widehat{y_0}^2$ to $\alpha^2$ to perform the LRT.

In Figure 2a, we plot the membership advantage for $\gamma > 1$ using Eq. (4). Since the difference between the variances of the model's output when $m = 0$ and $m = 1$ increases with $\gamma$ for $\gamma > 2$

---

[1]We detail additional experimental details and the computational hardware in Appendix E.

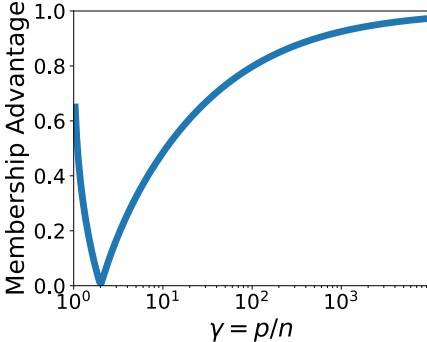
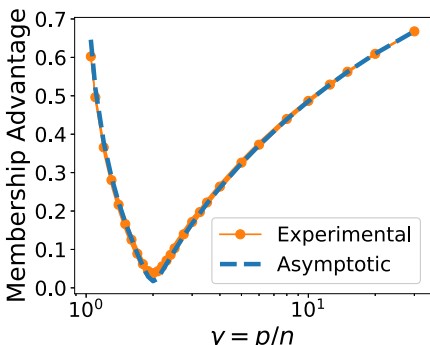

(a) Asymptotic membership advantage     (b) Empirical verification for non-asymptotic case

Figure 2: Increasing the number of parameters increases membership advantage. (a) Theoretical membership advantage for linear regression with Gaussian data (Eq. 4) as a function of the number of parameters for 100 sampled $x_0$'s, $n = 10^3$, $D = 10^7$. (b) We empirically approximate the membership advantage averaged over 20 sampled $x_0$'s for $n = 100$, $D = 3,000$, and $\sigma = 1$ by estimating the two posterior distributions for each $x_0$ using empirical histograms with 100,000 samples. We plot it alongside the theoretical asymptotic membership advantage, showing a close agreement between the two.

(cf., Fig. 1a), it becomes easier to distinguish the two distributions. This results in an increase in the membership advantage. The initial decrease in membership advantage when $\gamma \leq 2$ is a consequence of $\sigma_1 \leq \sigma_0$ in this regime as shown in Fig. 1a. Since $\sigma_0$ is decreasing in $p$, initially, this decrease makes the train and test output distributions harder to distinguish, leading to lower membership advantage and 0 advantage for $\gamma = 2$. However, for $\gamma > 2$, $\sigma_0$ decreases past $\sigma_1$ and membership advantage approaches 1 as $\gamma \to \infty$. In practice, the $\gamma < 2$ regime (only slightly overparameterized) is less interesting since models suffer larger generalization error in this setting (cf., Theorem 1 of [23]) and are rarely (if ever) used in practice. A key takeaway is that for linear regression in the Gaussian data setting, extreme overparameterization increases the vulnerability of a machine learning model to MI.

While Theorem 3.2 operates in the asymptotic regime, we empirically approximate membership advantage for $n = 100$ and $D = 3,000$ by approximating the posterior distributions with histograms of samples. As we see in Figure 2b, the asymptotic formula agrees very closely with the non-asymptotic experiment. Experimental details are provided in Appendix E.

## 4 Mitigating Membership Inference Attacks

Next, we extend our analysis from the previous section towards two methods commonly used for preserving privacy: regularization and noise addition. We present two key results. First, we show that for the overparameterized Gaussian data setting, ridge regularization actually *increases* membership advantage and is thus detrimental to privacy. Second, we show that the privacy-utility trade-off induced by reducing the number of parameters of a linear regression model with Gaussian data is equivalent to that of adding independent Gaussian noise to the output of a model that uses all available features.

### 4.1 Ridge-Regularized Linear Regression

We analyze membership inference in *ridge-regularized* linear regression in the same setting as Section 3.2 except that, now, the estimate is $\hat{\beta}_\lambda = (X_p^\top X_p + n\lambda I_p)^\dagger X_p^\top y$, where $\lambda$ is a regularization parameter. Larger values of $\lambda$ yield greater regularization, and $\lambda \to 0$ reduces to the case of Section 3.2. Regularization is a common method to reduce overfitting and has thus been proposed in previous works as a defense mechanism to protect models from MI attacks [13, 30].

A surprising observation resulting from our analysis is that, in the highly overparameterized regime ($\gamma \gg 1$), ridge regularization actually *increases* the model's vulnerability to MI attacks for linear

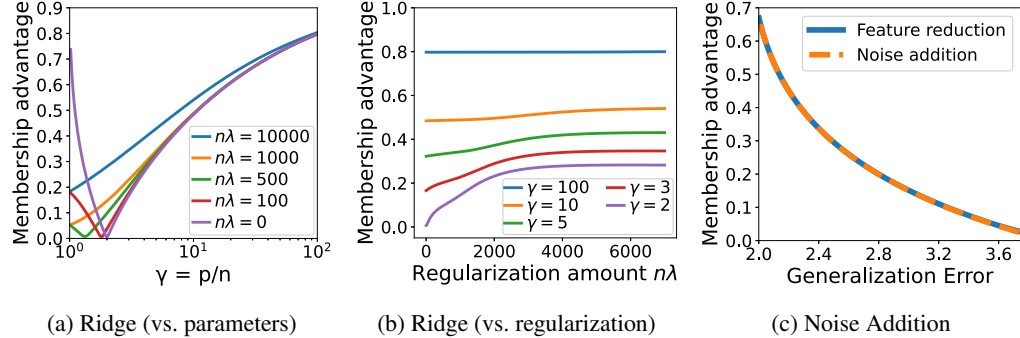

|  (a) Ridge (vs. parameters) | (b) Ridge (vs. regularization) | (c) Noise Addition |

Figure 3: We analyze ridge regression and noise addition, two different methods that aim to reduce membership advantage (MA). (a) MA vs. overparamterization $\gamma$ for a few ridge regularization strengths $\lambda$. Stronger ridge regularization harms privacy (increased MA). (b) MA vs. $\lambda$ for a few $\gamma$'s, explicitly showing how increasing $\lambda$ increases membership advantage. (c) We plot the privacy-utility trade-off obtained when tuning the number of parameters (feature reduction) and when adding independent noise to the output of a model that uses all available parameters (noise addition) and demonstrate that the two trade-offs are essentially equivalent. For (a) and (b), $n = 10^3$, $D = 10^7$, and $\sigma = 1$. For (c), $n = 100$, $D = 3000$, and $\sigma = 1$.

regression with Gaussian data. In Theorem D.1, presented in the appendix, we derive an analogous result for the distributions of the ridge-regularized predictions as Theorem 3.2 demonstrated for the unregularized case. In particular, if $A_\lambda^*$ is the adversary for the $\lambda$-regularized problem, its membership advantage $\text{Adv}(A_\lambda^*)$ is an increasing function in $\lambda$ when $\gamma \gg 1$. To visualize this, we plot the theoretical membership advantages for 100 sampled $x_0$'s (each with iid standard normal elements) for differing regularization strengths in Figure 3 with the same setting as in Figure 2a. In Figure 3a, we plot membership advantage as a function of the overparameterization ratio $\gamma$ for a few $n\lambda$ values. Conversely, In Figure 3b we plot membership advantage versus regularization $n\lambda$ for a few different values of $\gamma$. In particular, note how in both subplots, increasing $\lambda$ never decreases the membership advantage. This observation has been made empirically (but not analytically) for techniques that have similar regularizing effects, such as dropout [15], ensembling [22], and weight decay [30].

Intuitively, the reason ridge regularization increases MI vulnerability is because while it decreases the variance of the model's output on training points, it also significantly decreases the variance for test points such that the two output distributions become more distinguishable. In Figure 5 in the appendix, we plot the gap between the variances for the $m = 0$ and $m = 1$ cases for different regularization amounts $\lambda$ to visualize this effect.

## 4.2 Noise Addition vs. Feature Reduction

A consequence of Thm. 3.2 is that, in the Gaussian data setting, one can reduce vulnerability to MI attacks by simply *decreasing* the number of parameters/features. However, this comes at the cost of decreased generalization performance (utility) due to the "double descent" effect [6, 23, 7], wherein generalization error decreases with increased overparameterization. An alternative and popular method to increase a model's privacy is adding independent noise to the model output [24, 31], but this also decreases generalization performance. Interestingly, we show in this subsection that for Gaussian data, the privacy-utility trade-offs induced by both feature reduction and noise addition are actually equivalent.

**Feature reduction:** To characterize the privacy-utility trade-off of feature reduction, we first need the generalization error for a given number of parameters, provided in Corollary 2.2 of [23]:

**Proposition 4.1.** *(Adapted from Corollary 2.2 of [23]). In the same setting as Theorem 3.2, for $x_0 \sim \mathcal{N}(0, I_D)$, the generalization error is given by*

$$\mathbb{E}[(y - \hat{\beta}^\top x_{0,p})^2] = 1 + \sigma^2 + n \left( \frac{1 + \sigma^2 - \frac{p}{D}}{p - n - 1} - \frac{1}{D} \right).$$

Alternatively, this expression can be written as $1 + \sigma^2 + \mathbb{E}_{x_0}[\sigma_0^2] - 2\frac{n}{D}$, with $\sigma_0^2$ as in Theorem 3.2.

**Noise addition:** Next, we consider noise addition performed by perturbing the $m = 0$ model output with independent noise before releasing it: $\widehat{y_0} = \boldsymbol{x}_{0,p}^\top \hat{\beta} + \bar{\epsilon}$ where $\bar{\epsilon} \sim \mathcal{N}(0, \bar{\sigma}^2)$. Recall that MI is possible for overparameterized linear regression models because the variance of the output of test points ($m = 0$) is lower than the variance of the output for training points ($m = 1$) (Fig. 1). Thus, adding random noise to the model output when a test point $\boldsymbol{x}_0$ is queried can make the output distributions harder to distinguish. In the following lemma, we compute the membership advantage and generalization error of noise addition using the full set of features $p = D$.

**Lemma 4.2.** *Consider the setup of Theorem 3.2 restricted to $p = D$, $\sigma_1 > \sigma_0$, and $\gamma > 1$, and suppose independent $0$-mean, variance $\bar{\sigma}^2$ Gaussian noise is added to the outputs when $m = 0$ ($\boldsymbol{x}_0$ is not a member of the training set). Then, $(\widehat{y_0} \mid m = 0, \boldsymbol{x}_0) \sim \mathcal{N}(0, \sigma_0^2 + \bar{\sigma}^2)$. The optimal adversary is*

$$A_{\bar{\sigma}}^*(\boldsymbol{x}_0, \widehat{y_0}) := \mathbb{1}\left[\hat{y_0}^2 > \alpha_{\bar{\sigma}}^2\right] \quad where \quad \alpha_{\bar{\sigma}} := \sqrt{\frac{(\sigma_0^2 + \bar{\sigma}^2)\sigma_1^2 \log\left(\frac{\sigma_1^2}{(\sigma_0^2 + \bar{\sigma}^2)}\right)}{\sigma_1^2 - (\sigma_0^2 + \bar{\sigma}^2)}}.$$

*Its membership inference advantage is*

$$Adv(A_{\bar{\sigma}}^*) = \mathbb{E}_{\boldsymbol{x}_0}\left[2\left\{\Phi\left(\frac{\alpha_{\bar{\sigma}}}{\sqrt{\sigma_0^2 + \bar{\sigma}^2}}\right) - \Phi\left(\frac{\alpha_{\bar{\sigma}}}{\sigma_1}\right)\right\}\right]. \quad (5)$$

*Additionally, for $\boldsymbol{x}_0 \sim \mathcal{N}(0, \boldsymbol{I}_D)$, the generalization error incurred is*

$$\mathbb{E}[(y - \hat{\beta}^\top \boldsymbol{x}_0)^2] = 1 + \sigma^2 + n\left(\frac{\sigma^2}{D - n - 1} - \frac{1}{D}\right) + \bar{\sigma}^2. \quad (6)$$

The terms in Lemma 4.2 are similar to those in Theorem 3.2 except for the added dependence on the added noise's variance $\bar{\sigma}^2$ in the case that $m = 0$ ($\boldsymbol{x}_0$ is not a member). Increasing noise variance $\bar{\sigma}^2$ decreases membership advantage (Eq. (5)) at the cost of increased generalization error (Eq. (6)). Indeed, it is possible to add sufficient noise such that $\sigma_1^2 = \sigma_0^2 + \bar{\sigma}^2$, rendering the membership advantage 0, though possibly at the cost of impermissible generalization performance.

In Figure 3c, we plot the membership advantage vs. generalization error trade-offs for both feature reduction (blue) and noise addition (orange). The plots follow the setting of Fig. 2b. For the blue curve, we employ the expressions in Thm 3.2 and Prop 4.1 while varying the overparameterization ratio $\gamma$. For the orange curve, we use Lemma 4.2, using all available features ($p = D$) while varying the noise variance $\sigma^2$. Fig. 3c shows that the trade-offs induced by both feature reduction and noise addition are essentially equivalent.

# 5 More Complex Models

In this section, we present three more complex data models wherein we empirically observe increased overparameterization leading to increased MI vulnerability.

For each data model, we perform the following experiment. We first sample a $\boldsymbol{x}_0$ vector, which is the data point we wish to perform MI on. Then we sample a training dataset $\boldsymbol{X}$, measurements $\boldsymbol{y}$, as well as other random elements according to the data model. To obtain an $m = 0$ prediction, we learn a model on $\boldsymbol{X}$ that we then apply on $\boldsymbol{x}_0$. To obtain an $m = 1$ prediction, we first replace a row of $\boldsymbol{X}$ with $\boldsymbol{x}_0$ and the corresponding element of $\boldsymbol{y}$ with $y_0$ before learning the model and applying it on $\boldsymbol{x}_0$. Keeping $\boldsymbol{x}_0$ fixed throughout the experiment and resampling all other random data (such as $\boldsymbol{X}$) many times, we collect a large set of $m = 0$ and $m = 1$ prediction samples. We build a histogram of these samples by assigning them into fine discrete bins to obtain approximations of the conditional densities $\hat{P}(\widehat{y_0} \mid m = 1, \boldsymbol{x}_0)$ and $\hat{P}(\widehat{y_0} \mid m = 0, \boldsymbol{x}_0)$ needed for the optimal adversary (cf., Prop 3.1). To approximate membership advantage, we sum up the differences $\hat{P}(\widehat{y_0} \mid m_1, \boldsymbol{x}_0) - \hat{P}(\widehat{y_0} \mid m_0, \boldsymbol{x}_0)$ over all the histogram bins where $\hat{P}(\widehat{y_0} \mid m_1, \boldsymbol{x}_0) > \hat{P}(\widehat{y_0} \mid m_0, \boldsymbol{x}_0)$. We repeat this experiment 20 times, each with a newly sampled $\boldsymbol{x}_0$, and plot the means (as points) and the estimated standard errors (as shaded error regions) of the membership advantage values across the 20 experiments in Figure 4. We next discuss the data models in detail.

**Linear Regression on Latent Space Model:** We first consider the latent space model from [3], where the output variable $y_i$ is a noisy linear function of a data point's $d$ latent features $\boldsymbol{z}_i$, but one

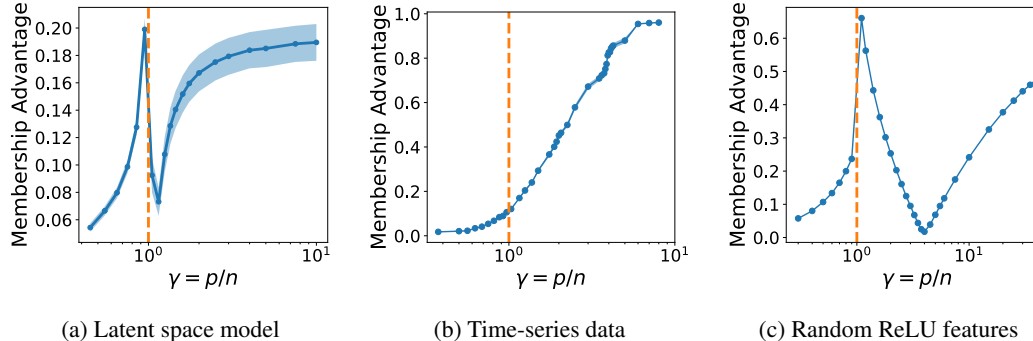

| (a) Latent space model | (b) Time-series data | (c) Random ReLU features |

Figure 4: Empirically measured membership advantage vs. parameterization for various data models detailed in Section 5. The dashed line at $\gamma = 1$ divides the underparameterized and overparameterized regions. For all settings, when sufficiently overparameterized, increasing the number of parameters increases vulnerability to membership inference attacks. (a) Latent space model with $n = 200$ where $p$ covariates of $d = 20$ latent features are observed. (b) Regression over $n = 128$ time samples of a linear combination of $D = 1024$ Fourier features. (c) Nonlinear random ReLU features with $n = 100$, $D = 5000$, and $\sigma = 1$.

only observes a vector $\boldsymbol{x}_i$ containing $p \geq d$ covariates rather than the direct features $\boldsymbol{z}_i$. Let $\boldsymbol{Z}$ be an $n \times d$ matrix where each row is the vector of latent features for an observation. We then have:

$$\boldsymbol{y} = \boldsymbol{Z}\beta + \epsilon, \qquad \boldsymbol{X}_{i,j} = \boldsymbol{w}_j^\top \boldsymbol{z}_i + u_{i,j}, \qquad \widehat{y}_0 = \boldsymbol{x}_0^\top \boldsymbol{X}_p^\dagger y,$$

where $\boldsymbol{w}_j$ is a $d$-dimensional vector, and $u_{i,j}$ is a noise term. In this experiment, we set $n = 200$, $d = 20$, and vary $p$. For each experiment, we sample a single $\boldsymbol{x}_0 \sim \mathcal{N}(0, \boldsymbol{I}_d)$ and a single set of $\boldsymbol{w}_j$ vectors, each from $\mathcal{N}(0, \boldsymbol{I}_d)$, and keep them fixed. We leave the other variables random with the following distributions: $\boldsymbol{z}_i \sim \mathcal{N}(0, \boldsymbol{I}_d)$, $\epsilon_i \sim \mathcal{N}(0, \sigma^2)$, $\beta \sim \mathcal{N}\left(0, \frac{1}{d}\boldsymbol{I}_d\right)$ and $u_{i,j} \sim \mathcal{N}(0, 1)$. In Fig. 4a, we plot the empirical membership advantage values, which increase with the number $p$ of features in the overparameterized regime.

**Noise-Free Time-Series Regression:** In this experiment, we consider a model that aims to interpolate a time-series signal using frequency components as the features. For example, consider a patient who visits a hospital at irregular times $t_i$ to get their blood glucose level measured. After obtaining a number of measurements taken over time, the hospital fits a time-series signal representing the patient's blood glucose level at any time. Using this learned model, an adversary wishes to identify whether the patient visited the hospital at a particular time $t_0$, that is, if $t_0$ is one such time point included in the hospital's training dataset for learning the patient's blood glucose level.

To formalize this, we fix $D = 1024$ as the number of frequency components each signal contains and let $M = 2D - 1$. Let $\boldsymbol{W}$ be the $D \times D$ matrix whose elements $\boldsymbol{W}_{kl} = \frac{1}{\sqrt{M+2}} \cos\left(\frac{2\pi kl}{M}\right)$. That is, each column of the matrix is half a period of an $M$-dimensional discrete cosine with frequency $\frac{2\pi k}{M}$. Sample a $\beta \sim \mathcal{N}\left(0, \frac{1}{D}\boldsymbol{I}_D\right)$, and let $\boldsymbol{z} = \boldsymbol{W}\beta$ denote the true length-$D$ signal. Thus, the signal is a random linear combination of cosines. We randomly select $n = 128$ indices from $1, 2, \ldots, D$ uniformly without replacement, and let $\boldsymbol{X}$ be the $n \times D$ matrix whose rows are the rows of $\boldsymbol{W}$ at these selected indices. Then, $\boldsymbol{y} = \boldsymbol{X}\beta$ is the signal observed at the randomly selected $n$ indices. The regressor learns $\hat{\beta} = \boldsymbol{X}_p^\dagger \boldsymbol{y}$ and then predicts the signal at any other time point $t_0$ as $\widehat{y}_0 = \boldsymbol{x}_{0,p}^\top \hat{\beta}$, where $\boldsymbol{x}_0$ is row $t_0$ of $\boldsymbol{W}$. Thus, identifying whether $t_0$ was a time point in the training dataset is equivalent to identifying if $\boldsymbol{x}_0$ was in $\boldsymbol{X}$. The membership advantage values for this task, plotted in Figure 4b, increases with the number $p$ of frequency components included in the model.

**Random ReLU Features:** We next consider a nonlinear data model based on Random ReLU feature networks [32, 33]. Let $\boldsymbol{Z}$ be a random $n \times D$ matrix whose elements are iid standard normal. Let $\boldsymbol{V}$ be a random $D \times p$ matrix whose rows are sampled iid from the surface of the unit sphere in $\mathbb{R}^p$. Let $\boldsymbol{X} = \max(\boldsymbol{Z}\boldsymbol{V}, 0)$, where the max is taken elementwise. The target variables are given by $\boldsymbol{y} = \boldsymbol{Z}\beta + \sigma\epsilon$, where $\beta \sim \mathcal{N}\left(0, \frac{1}{D}\boldsymbol{I}_D\right)$ and $\epsilon \sim \mathcal{N}(0, \boldsymbol{I}_n)$. Finally, for the data point $\boldsymbol{x}_0$, let its prediction be $\widehat{y}_0 = \boldsymbol{x}_0^\top \boldsymbol{X}^\dagger y$. We plot the membership advantage in Figure 4c with $n = 100$, $D = 5000$, and $\sigma = 1$. We again observe that in the highly overparametrized regime, membership advantage increases with parameters.

# 6 Discussion and Conclusions

We have shown theoretically for (regularized) linear regression with Gaussian data and empirically for more complex models (latent space regression, time-series regression using Fourier components, and random ReLU features) that increasing the number of model parameters renders them more vulnerable to membership inference attacks. Thus, while overparameterization may be attractive for its robust generalization performance, one must proceed with caution to ensure the learned model does not lead to unintended privacy leakages.

More speculatively, we hypothesize that the same overparameterization/vulnerability tradeoff should exist in many machine learning models (e.g., deep networks) beyond those we have studied. Intuitively, the output of a model that achieves zero training error but generalizes well must i) fit to any noise (e.g. additive Gaussian noise) present in the training data to get a perfect fit to the noisy training data but also ii) eliminate the effect of noise in the training data in predicting for unseen data to achieve good generalization. This disparate behavior towards training and non-training data points leads to different output distributions when the input is or is not among the training data and is universal for overparameterized models. Ultimately, this causes a difference in the distributions of training and test predictions that can be leveraged to perform a membership attack.

There are still many open questions in this line of research. While we have shown multiple settings where reducing the number of parameters can increase privacy, it remains to be verified that the phenomenon holds widely for other types of machine learning settings such as language tasks or large-scale image recognition. Another interesting next step would be to investigate how increased overparameterization affects privacy for models trained with privacy-preserving techniques or membership inference defense schemes other than ridge regularization. We believe the findings of our work can provide insights towards developing the next generation of privacy-preserving techniques. It is our hope that the observations and analyses in this paper take a step towards keeping sensitive training data safer in a world increasingly intertwined with machine learning.

**Acknowledgements**

This work was supported by NSF grants CCF-1911094, IIS-1838177, and IIS-1730574; ONR grants N00014-18-12571, N00014-20-1-2534, and MURI N00014-20-1-2787; AFOSR grant FA9550-22-1-0060; and a Vannevar Bush Faculty Fellowship, ONR grant N00014-18-1-2047.

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
