# A    Additional Discussions

## A.1    Limitations

In this work, we focus on the optimal membership inference adversary. We study this because of how it serves as an upper bound for all other attacks and because of how it yields interpretable and fundamental theoretical results. The optimal membership inference adversary has full knowledge of the learning model's output distributions when the data point of interest is a member or non-member of the training dataset. In practice, the adversary rarely has such full knowledge, and the learning model's output distributions have to be approximated using shadow models [13], or the entire attack has to be simplified, such as with a loss threshold [9, 14]. Our study does not analyze how our results are affected by the non-optimality of these more practical attacks.

## A.2    Ethical Considerations

It is the hope of the authors that by more clearly exposing the link between membership inference vulnerability and generalization performance, researchers can make informed decisions about how to achieve the best trade-off they can for their application. That said, by studying the performance of optimal membership inference attacks, it is possible that this work will call attention to vulnerabilities in existing model architectures which may be exploited. Furthermore, in settings where privacy is absolutely crucial, such as in medical applications, additional care should be taken to guard privacy beyond the guarantees of this work.

# B    Proofs

## B.1    Proof of Proposition 3.1

We first present the proof of the form of the optimal membership inference adversary given in Proposition 3.1.

*Proof of Proposition 3.1.* Conditioned on $m = 0$, we have that $(\boldsymbol{x}_0, y_0)$ is drawn from $\mathcal{D}$. Conditioned on $m = 1$, we have that $(\boldsymbol{x}_0, y_0)$ is an element chosen randomly from $S$, whose elements are themselves drawn from $\mathcal{D}$. Thus, in both the $m = 0$ and $m = 1$ cases, $x_0$ has the same distribution. We thus have:

$$
\begin{aligned}
A^* &= \arg\max_A \operatorname{Adv}(A) \\
&= \arg\max_A \mathbb{P}(A((\boldsymbol{x}_0, \widehat{y_0})) = 1 \mid m = 1) - \mathbb{P}(A((\boldsymbol{x}_0, \widehat{y_0})) = 1 \mid m = 0) \\
&= \arg\max_A \mathbb{E}_{\boldsymbol{x}_0}[\mathbb{P}(A((\boldsymbol{x}_0, \widehat{y_0})) = 1 \mid m = 1, \boldsymbol{x}_0) - \mathbb{P}(A((\boldsymbol{x}_0, \widehat{y_0})) = 1 \mid m = 0, \boldsymbol{x}_0)] \\
&= \arg\max_A \mathbb{E}_{\boldsymbol{x}_0}\left[\int_{\mathbb{R}} \mathbb{1}_{A(\boldsymbol{x}_0, \widehat{y_0})=1} \left(P(\widehat{y_0} \mid m = 1, \boldsymbol{x}_0) - P(\widehat{y_0} \mid m - 1, \boldsymbol{x}_0)\right) dP\right],
\end{aligned}
$$

where in the third line, the randomness over $\boldsymbol{x}_0$ is removed from the probability. To maximize the integral, we set $A(\boldsymbol{x}_0, \widehat{y_0}) = 1$ if $P(\widehat{y_0} \mid m = 1, \boldsymbol{x}_0) - P(\widehat{y_0} \mid m = 1, \boldsymbol{x}_0) > 0$ and 0 otherwise.    □

## B.2    Tools for Asymptotic Analysis

The following lemmas are used in the proofs of Theorems 3.2 and D.1. We begin with the following lemma, which is a generalized version of the Marchenko-Pastur theorem [34–36].

**Lemma B.1.** *Let $\boldsymbol{X}_n \in \mathbb{R}^{n \times p}$ be a sequence of random matrices with i.i.d. $\mathcal{N}(0, 1)$ entries. Consider the the sample covariance matrix $\widehat{\boldsymbol{\Sigma}} = (1/n)\boldsymbol{X}_n^\top \boldsymbol{X}_n$. Let $\boldsymbol{C}_n \in \mathbb{R}^{p \times p}$ be a sequence of matrices such that $\operatorname{Tr}(\boldsymbol{C}_n)$ is uniformly bounded with probability one. As $n, p \to \infty$ with $p/n = \gamma \in (0, \infty)$, it holds that almost surely,*

$$
\operatorname{Tr}\left(\boldsymbol{C}_n\left((\boldsymbol{\Sigma} + \lambda \boldsymbol{I}_p)^{-1} - g(-\lambda)\boldsymbol{I}_p\right)\right) \to 0, \qquad \operatorname{Tr}\left(\boldsymbol{C}_n\left((\boldsymbol{\Sigma} + \lambda \boldsymbol{I}_p)^{-2} - g'(-\lambda)\boldsymbol{I}_p\right)\right) \to 0
$$

*where $g(\lambda)$ is the Stieltjes transform of the Marchenko-Pastur law with parameter $\gamma$.*

We use the following Lemma in computing the asymptotic distribution of the output.

**Lemma B.2.** *Let $\boldsymbol{y}_n \in \mathbb{R}^n$ be a sequence of i.i.d. $\mathcal{N}(0, \boldsymbol{I}_n)$ random vectors. Also, let $\boldsymbol{x}_n \in \mathbb{R}^n$ be a sequence of random vectors with spherically symmetric distribution such that $\|\boldsymbol{x}_n\|_2 \xrightarrow{a.s.} \sigma$. Further, assume that $\boldsymbol{x}_n, \boldsymbol{y}_n$ are independent. Then $\boldsymbol{x}_n^\top \boldsymbol{y}$ converges weakly to a zero mean gaussian with variance $\sigma^2$.*

*Proof.* We can write

$$\boldsymbol{x}_n^\top \boldsymbol{y}_n = \|\boldsymbol{x}_n\|_2 \left( \frac{\boldsymbol{x}_n}{\|\boldsymbol{x}_n\|} \right)^\top \boldsymbol{y}_n = \|\boldsymbol{x}_n\|_2 \, \boldsymbol{u}_n^\top \boldsymbol{y}_n$$

where $\boldsymbol{u}_n \in S^{n-1}$ is uniformly distributed over the unit sphere and is independent from $\boldsymbol{y}_n$. Therefore, we can fix $\boldsymbol{u}_n$ to be the first standard unit vector and the distribution of $\boldsymbol{x}_n^\top \boldsymbol{y}_n$ is the same as $\|\boldsymbol{x}_n\|_2 \, y_{n,1}$ where $y_{n,1} \sim \mathcal{N}(0,1)$. Hence, using $\|\boldsymbol{x}_n\|_2 \xrightarrow{a.s.} \sigma$, we deduce the result. $\square$

## B.3  Proof of Theorem 3.2

*Proof of Theorem 3.2.* Let $\boldsymbol{X}_{\overline{p}}$ denote the matrix formed by removing the first $p$ columns from $\boldsymbol{X}$, and let $\beta_{\overline{p}}$ denote the vector formed by removing the first $p$ elements from $\beta$. Recall that

$$
\begin{aligned}
(\widehat{y}_0 \mid m = 0) &= \boldsymbol{x}_0^\top \boldsymbol{X}_p^\dagger (\boldsymbol{X}\beta + \epsilon) \\
&= \boldsymbol{x}_0^\top \boldsymbol{X}_p^\dagger (\boldsymbol{X}_p \beta_p + \eta)
\end{aligned}
$$

where $\eta = \boldsymbol{X}_{\overline{p}} \boldsymbol{\beta}_{\overline{p}} + \epsilon \sim \mathcal{N}\left(0, \left(1 + \sigma^2 - \frac{p}{D}\right) \boldsymbol{I}_n\right)$. First note that the distributions of $\boldsymbol{X}_p^\top \boldsymbol{X}_p^\dagger \boldsymbol{x}_0$ are $\boldsymbol{X}_p^\dagger \boldsymbol{x}_0$ are spherically symmetric and letting $\widehat{\boldsymbol{\Sigma}} \triangleq (1/n) \boldsymbol{X}_p^\top \boldsymbol{X}_p$ and $\boldsymbol{P}$ to be orthogonal projection onto row space of $\boldsymbol{X}_p$ we have

$$
\begin{aligned}
\frac{1}{D} \|\boldsymbol{X}_p^\top \boldsymbol{X}_p^\dagger \boldsymbol{x}_0\|_2^2 = \frac{1}{D} \|\boldsymbol{P}\boldsymbol{x}_0\|_2^2 &= \frac{1}{D} \lim_{\lambda \to 0} \boldsymbol{x}_0^\top \left(\widehat{\boldsymbol{\Sigma}} + \lambda \boldsymbol{I}_p\right)^{-1} \widehat{\boldsymbol{\Sigma}} \boldsymbol{x}_0 \\
&= \frac{1}{D} \|\boldsymbol{x}_0\|_2^2 - \frac{1}{D} \lim_{\lambda \to 0} \lambda \boldsymbol{x}_0^\top \left(\widehat{\boldsymbol{\Sigma}} + \lambda \boldsymbol{I}_p\right)^{-1} \boldsymbol{x}_0,
\end{aligned}
$$

and,

$$
\begin{aligned}
\|\boldsymbol{X}_p^\dagger \boldsymbol{x}_0\|_2^2 &= \frac{1}{n} \lim_{\lambda \to 0} \boldsymbol{x}_0^\top \left(\widehat{\boldsymbol{\Sigma}} + \lambda \boldsymbol{I}_p\right)^{-1} \widehat{\boldsymbol{\Sigma}} \left(\widehat{\boldsymbol{\Sigma}} + \lambda \boldsymbol{I}_p\right)^{-1} \boldsymbol{x}_0 \\
&= \frac{1}{n} \lim_{\lambda \to 0} \boldsymbol{x}_0^\top \left[ \left(\widehat{\boldsymbol{\Sigma}} + \lambda \boldsymbol{I}_p\right)^{-1} - \lambda \left(\widehat{\boldsymbol{\Sigma}} + \lambda \boldsymbol{I}_p\right)^{-2} \right] \boldsymbol{x}_0^\top.
\end{aligned}
$$

Thus, using Lemma B.2, both $\frac{1}{D} \|\boldsymbol{X}_p^\top \boldsymbol{X}_p^\dagger \boldsymbol{x}_0\|_2^2$ and $\|\boldsymbol{X}_p^\dagger \boldsymbol{x}_0\|_2^2$ converge to a fixed limit as $n \to \infty$, almost surely. Therefore, using Lemma B.2, $\widehat{y}_0$ converges weakly to a gaussian. Now, we compute its variance. Since $\eta$ and $\beta$ are both zero-mean independent Gaussians and are thus orthogonal in expectation, we have by the Pythagorean theorem:

$$\mathbb{E}\left[\widehat{y}_0^2 \mid m = 0\right] = \mathbb{E}\left[(\boldsymbol{x}_0^\top \boldsymbol{X}_p^\dagger \boldsymbol{X}_p \beta_p)^2\right] + \mathbb{E}\left[(\boldsymbol{x}_0^\top \boldsymbol{X}_p^\dagger \eta)^2\right].$$

We start with the first term.

Note that since, $p > n$, $\boldsymbol{X}_p$ does not have linearly independent columns. Let $\boldsymbol{P} = \boldsymbol{X}_p^\dagger \boldsymbol{X}_p$. We have:

$$
\begin{aligned}
\mathbb{E}\left[(\boldsymbol{x}_0^\top \boldsymbol{X}_p^\dagger \boldsymbol{X}_p \beta_p)^2\right] &= \mathbb{E}\left[\mathrm{Tr}\left(\boldsymbol{x}_0^\top \boldsymbol{P}\beta_p\beta_p^\top \boldsymbol{P}^\top \boldsymbol{x}_0\right)\right] \\
&= \mathbb{E}\left[\mathrm{Tr}\left(\beta_p\beta_p^\top \boldsymbol{P}^\top \boldsymbol{x}_0\boldsymbol{x}_0^\top \boldsymbol{P}\right)\right] \\
&= \mathrm{Tr}\left(\mathbb{E}\left[\beta_p\beta_p^\top \boldsymbol{P}^\top \boldsymbol{x}_0\boldsymbol{x}_0^\top \boldsymbol{P}\right]\right) \\
&= \mathrm{Tr}\left(\mathbb{E}\left[\beta_p\beta_p^\top\right]\mathbb{E}\left[\boldsymbol{P}^\top \boldsymbol{x}_0\boldsymbol{x}_0^\top \boldsymbol{P}\right]\right) \\
&= \frac{1}{D}\mathrm{Tr}\left(\mathbb{E}\left[\boldsymbol{P}^\top \boldsymbol{x}_0\boldsymbol{x}_0^\top \boldsymbol{P}\right]\right) \\
&= \frac{1}{D}\mathbb{E}\left[\|\boldsymbol{P}^\top \boldsymbol{x}_0\|^2\right] \\
&= \frac{1}{D}\frac{n}{p}\|\boldsymbol{x}_0\|^2.
\end{aligned}
$$

In the last line, we use the same argument as in Section 2.2 of [23], using the facts that $\boldsymbol{P}$ is the orthogonal projection to the row space of $\boldsymbol{X}_p$ and that the Gaussian distribution is invariant to rotations.

We now consider the second term:

$$
\begin{aligned}
\mathbb{E}\left[(\boldsymbol{x}_0^\top \boldsymbol{X}_p^\dagger \eta)^2\right] &= \left(1 + \sigma^2 - \frac{p}{D}\right)\boldsymbol{x}_0^\top \mathbb{E}\left[\boldsymbol{X}_p^\dagger \boldsymbol{X}_p^{\dagger\top}\right]\boldsymbol{x}_0 \\
&= \left(1 + \sigma^2 - \frac{p}{D}\right)\boldsymbol{x}_0^\top \mathbb{E}\left[\left(\boldsymbol{X}_p^\top \boldsymbol{X}_p\right)^\dagger \boldsymbol{X}_p^\top \boldsymbol{X}_p\left(\boldsymbol{X}_p^\top \boldsymbol{X}_p\right)^{\dagger\top}\right]\boldsymbol{x}_0 \\
&= \left(1 + \sigma^2 - \frac{p}{D}\right)\boldsymbol{x}_0^\top \mathbb{E}\left[\left(\boldsymbol{X}_p^\top \boldsymbol{X}_p\right)^\dagger\right]\boldsymbol{x}_0.
\end{aligned}
$$

where $\left(\boldsymbol{X}_p^\top \boldsymbol{X}_p\right)^\dagger$ has the generalized inverse Wishart distribution with expectation equal to $\mathbb{E}\left[\left(\boldsymbol{X}_p^\top \boldsymbol{X}_p\right)^\dagger\right] = \frac{n}{p}\frac{1}{p-n-1}\boldsymbol{I}_p$ (Theorem 2.1 in [37]). Thus, we have:

$$
\mathbb{E}\left[(\boldsymbol{x}_0^\top \boldsymbol{X}_p^\dagger \eta)^2\right] = \left(\frac{n}{p}\right)\left(\frac{1 + \sigma^2 - \frac{p}{D}}{p-n-1}\right)\|\boldsymbol{x}_0\|^2
$$

Adding this with the result for the first term gives the desired result. When $m = 1$, since we are in the overparameterized regime, $\boldsymbol{X}_p$ is a fat matrix. Thus, the regressor memorizes the training data and the training error is equal to zero. $\boldsymbol{x}_0$ is part of training set, and so $\widehat{y_0} = \boldsymbol{x}_0^\top \beta + \epsilon$. Since $\beta \sim \mathcal{N}\left(0, \frac{1}{D}\boldsymbol{I}_p\right)$, we have that $\boldsymbol{x}_0^\top \beta \sim \mathcal{N}(0, \frac{1}{D}\|\boldsymbol{x}_0\|^2)$. Since $\epsilon \sim \mathcal{N}(0, \sigma^2)$, we have that $\widehat{y_0} = \boldsymbol{x}_0^\top \beta + \epsilon \sim \mathcal{N}(0, \frac{1}{D}\|\boldsymbol{x}_0\|^2 + \sigma^2)$.

The probability distribution functions of the two Gaussians are then equal at $\pm\alpha$:

$$\frac{1}{\sigma_0\sqrt{2\pi}}\exp\left(-\frac{1}{2}\left(\frac{\alpha}{\sigma_0}\right)^2\right) = \frac{1}{\sigma_1\sqrt{2\pi}}\exp\left(-\frac{1}{2}\left(\frac{\alpha}{\sigma_1}\right)^2\right)$$

$$\frac{\sigma_1}{\sigma_0} = \exp\left(-\frac{1}{2}\left(\left(\frac{\alpha}{\sigma_1}\right)^2 - \left(\frac{\alpha}{\sigma_0}\right)^2\right)\right)$$

$$\frac{\sigma_1}{\sigma_0} = \exp\left(-\frac{\alpha^2}{2}\frac{\sigma_0^2 - \sigma_1^2}{\sigma_0^2\sigma_1^2}\right)$$

$$\log\left(\frac{\sigma_1}{\sigma_0}\right) = -\frac{\alpha^2}{2}\frac{\sigma_0^2 - \sigma_1^2}{\sigma_0^2\sigma_1^2}$$

$$\alpha = \sqrt{\frac{2\sigma_0^2\sigma_1^2\log\left(\frac{\sigma_1}{\sigma_0}\right)}{\sigma_1^2 - \sigma_0^2}}$$

$$\alpha = \sqrt{\frac{\sigma_0^2\sigma_1^2\log\left(\frac{\sigma_1^2}{\sigma_0^2}\right)}{\sigma_1^2 - \sigma_0^2}}.$$

The membership advantage is then derived by writing out the probabilities in Definition 2.1 in terms of the Gaussian cumulative distribution functions, noting that the decision region switches at $\pm\alpha$. $\square$

*Proof of Lemma 4.2.* The lemma follows identically to Theorem 3.2 with an additional additive $\bar{\sigma}^2$ to $\sigma_0^2$ due to the noise added in the $m = 0$ case. The remainder follows by plugging in $p = D$ and applying Prop. 4.1 for the generalization error. $\square$

## C  Posterior Distributions in Non-Asymptotic Regime

Let $f_{a|b}$ denote the probability density function of a random variable $a$ conditioned on $b$. The following lemma derives the non-asymptotic probability densities of the prediction output of minimum norm least squares, conditioned on the $m = 0$ and $m = 1$ events and the choice of test point $\boldsymbol{x}_0$. For a matrix $\boldsymbol{X} \in \mathbb{R}^{n\times D}$ and $p \leq D$, let $\boldsymbol{X}_p$ denote the submatrix of the first $p$ columns of $\boldsymbol{X}$. For a vector $\boldsymbol{x} \in \mathbb{R}^D$, let $\boldsymbol{x}_p \in \mathbb{R}^p$ be defined accordingly.

**Lemma C.1.** *Let $\hat{\beta}$ denote the minimum norm least squares interpolator computed from a random design matrix $\boldsymbol{X} \in \mathbb{R}^{n\times D}$ and data $\boldsymbol{y}$. Conditioned on $n < p \leq D$ and on $\boldsymbol{x}_0$, we have that $\boldsymbol{x}_{0,p}^\top\hat{\beta} \mid \boldsymbol{x}_0, \{m = 1\} \sim \mathcal{N}\left(0, \sigma_1^2\right)$, where $\sigma_1$ is defined as in Theorem 3.2. Furthermore,*

$$f_{\boldsymbol{x}_{0,p}^\top\hat{\beta}|\boldsymbol{x}_0,\{m=0\}}(x)$$

$$= D^{\frac{D}{2}} \int_{\mathbb{R}^{n\times D}} \int_{\mathbb{R}^D} \frac{\exp\left[-\frac{1}{2}\left[\left(\frac{x - \boldsymbol{x}_{0,p}^\top\boldsymbol{X}_p^\top(\boldsymbol{X}_p\boldsymbol{X}_p^\top)^{-1}\boldsymbol{X}\beta}{\sigma\|\boldsymbol{x}_{0,p}\|_{\boldsymbol{X}_p(\boldsymbol{X}_p\boldsymbol{X}_p^\top)^{-2}\boldsymbol{X}_p}}\right)^2 + D\beta^\top\beta + \mathrm{Tr}\left(\boldsymbol{X}^\top\boldsymbol{X}\right)\right]\right]}{\sigma(2\pi)^{\frac{nD+D+1}{2}}\|\boldsymbol{x}_{0,p}\|_{\boldsymbol{X}_p(\boldsymbol{X}_p\boldsymbol{X}_p^\top)^{-2}\boldsymbol{X}_p}} d\beta d\boldsymbol{X}.$$

*Remark C.2.* While the density of $\boldsymbol{x}_{0,p}^\top\hat{\beta} \mid \boldsymbol{x}_0, \{m = 0\}$ cannot be written in a closed form, one may easily sample according to it, by first, sampling random $\boldsymbol{X}, \beta$ and then computing the minimum norm least squares interpolator.

*Proof of Lemma C.1.* Recall that conditioned on the design matrix $\boldsymbol{X}$ and true coefficients $\beta$, the labels $\boldsymbol{y}$ follow $\boldsymbol{y} \mid \boldsymbol{X}, \beta \sim \mathcal{N}(\boldsymbol{X}\beta, \sigma^2\boldsymbol{I}_n)$. Then, the minimum norm least squares solution $\hat{\beta}$ using the first $p$ features follows

$$\hat{\beta} \mid \boldsymbol{X}, \beta \sim \mathcal{N}\left(\boldsymbol{X}_p^\top(\boldsymbol{X}_p\boldsymbol{X}_p^\top)^{-1}\boldsymbol{X}\beta, \sigma^2\boldsymbol{X}_p^\top(\boldsymbol{X}_p\boldsymbol{X}_p^\top)^{-2}\boldsymbol{X}_p\right).$$

Hence, for the $m = 0$ case where a fresh point $\boldsymbol{x}_0$ is sampled, we have that the distribution of the model output conditioned on the design matrix $\boldsymbol{X}$ and true coefficients $\beta$ is

$$\boldsymbol{x}_{0,p}^\top\hat{\beta} \mid \boldsymbol{X}, \beta, \boldsymbol{x}_0, \{m = 0\} \sim \mathcal{N}\left(\boldsymbol{x}_{0,p}^\top\boldsymbol{X}_p^\top(\boldsymbol{X}_p\boldsymbol{X}_p^\top)^{-1}\boldsymbol{X}\beta, \sigma^2\|\boldsymbol{x}_{0,p}\|^2_{\boldsymbol{X}_p^\top(\boldsymbol{X}_p\boldsymbol{X}_p^\top)^{-2}\boldsymbol{X}_p}\right)$$

for $\|x\|_A := \sqrt{x^\top A x}$ for any semidefinite matrix $A$ where we have additionally conditioned over any randomness in the choice of $x_0$.

In the $m = 1$ case, where $x_0$ is sampled uniformly from the rows of $X$, we have that $x_{0,p}^\top \hat{\beta} = y_0 = x_0^\top \beta + \epsilon$, the associated label for $x_0$ since the linear regressor interpolates the training data. Hence

$$x_{0,p}^\top \hat{\beta} \mid X, \beta, x_0, \{m = 1\} \equiv x_{0,p}^\top \hat{\beta} \mid x_0, \{m = 1\} \sim \mathcal{N}\left(0, \frac{\|x_0\|^2}{D} + \sigma^2\right)$$

Let $f_{x_{0,p}^\top \hat{\beta} \mid X, \beta, x_0, \{m=0\}}$ denote the pdf of the random variable $x_p^\top \hat{\beta} \mid X, \beta, x_0, \{m = 0\}$ and $f_{x_{0,p}^\top \hat{\beta} \mid X, \beta, x_0, \{m=1\}}$ be defined similarly. Let $f_X$ denote the density of $X$, a standard matrix-normal random variable, and let $f_\beta$ denote the density of $\beta \sim \mathcal{N}\left(0, \frac{1}{D}I_D\right)$. Then, we have that

$$f_{x_{0,p}^\top \hat{\beta} \mid x_0, \{m=0\}}(x)$$
$$= \int_{\mathbb{R}^{n \times D}} \int_{\mathbb{R}^D} f_{x_{0,p}^\top \hat{\beta} \mid X, \beta, x_0, \{m=0\}} f_\beta f_X \, d\beta dX$$
$$= D^{\frac{D}{2}} \int_{\mathbb{R}^{n \times D}} \int_{\mathbb{R}^D} \frac{\exp\left[-\frac{1}{2}\left[\left(\frac{x - x_{0,p}^\top X_p^\top (X_p X_p^\top)^{-1} X \beta}{\sigma \|x_{0,p}\|_{X_p(X_p X_p^\top)^{-2} X_p}}\right)^2 + D\beta^\top \beta + \mathrm{Tr}\left(X^\top X\right)\right]\right]}{\sigma(2\pi)^{\frac{nD+D+1}{2}} \|x_{0,p}\|_{X_p(X_p X_p^\top)^{-2} X_p}} \, d\beta dX.$$

$\square$

**Lemma C.3.** *Let $\hat{\beta}_\lambda = (X_p^\top X_p + n\lambda I)^{-1} X_p^\top y$ denote ridge regularized least squares estimator computed from random design matrix $X \in \mathbb{R}^{n \times D}$, data $y$, and subset of first $p$ features. Conditioned on the choice of test point $x_0$, we have that in the $m = 0$ case, where a fresh test point is drawn from the data distribution,*

$$f_{x_{0,p}^\top \hat{\beta} \mid x_0, \{m=0\}}(x) = \frac{D^{\frac{D}{2}}}{\sigma(2\pi)^{\frac{nD+D+1}{2}}}$$
$$\times \int_{\mathbb{R}^{n \times D}} \int_{\mathbb{R}^D} \frac{\exp\left[-\frac{1}{2}\left[\left(\frac{x - x_{0,p}^\top (X_p^\top X_p + n\lambda I)^{-1} X_p^\top X \beta}{\sigma \|x_{0,p}\|_{(X_p^\top X_p + n\lambda I)^{-1} X_p^\top X_p (X_p^\top X_p + n\lambda I)^{-1}}}\right)^2 + D\beta^\top \beta + \mathrm{Tr}\left(X^\top X\right)\right]\right]}{\|x_{0,p}\|_{(X_p^\top X_p + n\lambda I)^{-1} X_p^\top X_p (X_p^\top X_p + n\lambda I)^{-1}}} \, d\beta dX.$$

*Furthermore, conditioned on $m = 1$ when $x_0$ is a row of $X$ we have that*

$$f_{x_{0,p}^\top \hat{\beta} \mid x_0, \{m=1\}}(x) = \frac{D^{\frac{D}{2}}}{\sigma(2\pi)^{\frac{nD+1}{2}}}$$
$$\times \int_{\mathbb{R}^{(n-1) \times D}} \int_{\mathbb{R}^D} \frac{\exp\left[-\frac{1}{2}\left[\left(\frac{x - x_{0,p}^\top (X_p^\top X_p + n\lambda I)^{-1} X_p^\top X \beta}{\sigma \|x_{0,p}\|_{(X_p^\top X_p + n\lambda I)^{-1} X_p^\top X_p (X_p^\top X_p + n\lambda I)^{-1}}}\right)^2 + D\beta^\top \beta + \mathrm{Tr}\left(\tilde{X}^\top \tilde{X}\right)\right]\right]}{\|x_{0,p}\|_{(X_p^\top X_p + n\lambda I)^{-1} X_p^\top X_p (X_p^\top X_p + n\lambda I)^{-1}}} \, d\beta d\tilde{X}.$$

*where without loss of generality, we take $x_0$ to be the first row of $X$ and $\tilde{X}$ to denote the matrix of the final $n - 1$ rows of $X$.*

*Remark* C.4. As in the case of Lemma C.1, one can efficiently sample from the above distribution by first drawing a Gaussian random matrix $X$, the Gaussian random vector $\beta$, the Bernoulli random variable $m$, and then either a new test point $x_0$ or a row of $X$ and learning the ridge-regularized estimator $\hat{\beta}_\lambda$.

*Proof of Lemma C.3.* Note that conditioned on the design matrix $X$ and the true coefficients $\beta$, the labels $y$ follow $y \mid X, \beta \sim \mathcal{N}(X\beta, \sigma^2 I_n)$. Next, for

$$\hat{\beta}_\lambda := (X_p^\top X_p + n\lambda I_p)^{-1} X_p^\top y$$

the $\lambda$-ridge regularized estimator, we have that

$$\hat{\beta}_\lambda \mid \boldsymbol{X}, \beta \sim \mathcal{N}\left((\boldsymbol{X}_p^\top \boldsymbol{X}_p + n\lambda \boldsymbol{I}_p)^{-1} \boldsymbol{X}_p^\top \boldsymbol{X}\beta, \sigma^2 (\boldsymbol{X}_p^\top \boldsymbol{X}_p + n\lambda \boldsymbol{I}_p)^{-1} \boldsymbol{X}_p^\top \boldsymbol{X}_p (\boldsymbol{X}_p^\top \boldsymbol{X}_p + n\lambda \boldsymbol{I}_p)^{-1}\right).$$

Hence,

$$\boldsymbol{x}_{0,p}^\top \hat{\beta}_\lambda \mid \boldsymbol{X}, \boldsymbol{x}_0, \beta$$
$$\sim \mathcal{N}\left(\boldsymbol{x}_{0,p}^\top (\boldsymbol{X}_p^\top \boldsymbol{X}_p + n\lambda \boldsymbol{I}_p)^{-1} \boldsymbol{X}_p^\top \boldsymbol{X}\beta, \sigma^2 \boldsymbol{x}_{0,p}^\top (\boldsymbol{X}_p^\top \boldsymbol{X}_p + n\lambda \boldsymbol{I}_p)^{-1} \boldsymbol{X}_p^\top \boldsymbol{X}_p (\boldsymbol{X}_p^\top \boldsymbol{X}_p + n\lambda \boldsymbol{I}_p)^{-1} \boldsymbol{x}_{0,p}\right),$$

where we have additionally conditioned over any randomness in the choice of $\boldsymbol{x}_0$.

In the $m = 0$ case, where $\boldsymbol{x}_0$ is a freshly drawn point independent of the data $\boldsymbol{X}$, we may marginalize to remove the conditioning. Let $f_{\boldsymbol{x}_{0,p}^\top \hat{\beta}_\lambda \mid \boldsymbol{X}, \beta, \boldsymbol{x}_0, \{m=0\}}$ denote the probability density function of the random variable $\boldsymbol{x}_{0,p}^\top \hat{\beta}_\lambda \mid \boldsymbol{X}, \beta, \boldsymbol{x}_0, \{m = 0\}$ and $f_{\boldsymbol{x}_{0,p}^\top \hat{\beta} \mid \boldsymbol{X}, \boldsymbol{x}_0, \{m=1\}}$ be defined similarly. Let $f_{\boldsymbol{X}}$ denote the density of $\boldsymbol{X}$, a standard matrix-normal random variable, and let $f_\beta$ denote the density of $\beta \sim \mathcal{N}\left(0, \frac{1}{D}\boldsymbol{I}_D\right)$. Then we have that

$$f_{\boldsymbol{x}_{0,p}^\top \hat{\beta} \mid \boldsymbol{x}_0, \{m=0\}}(x) = \int_{\mathbb{R}^{n \times D}} \int_{\mathbb{R}^D} f_{\boldsymbol{x}_{0,p}^\top \hat{\beta} \mid \boldsymbol{X}, \boldsymbol{x}_0, \{m=0\}} f_\beta f_{\boldsymbol{X}} \, d\beta d\boldsymbol{X}.$$

Thus,

$$f_{\boldsymbol{x}_{0,p}^\top \hat{\beta} \mid \boldsymbol{x}_0, \{m=0\}}(x) = \frac{D^{\frac{D}{2}}}{\sigma(2\pi)^{\frac{nD+D+1}{2}}}$$
$$\times \int_{\mathbb{R}^{n \times D}} \int_{\mathbb{R}^D} \frac{\exp\left[-\frac{1}{2}\left[\left(\frac{x - \boldsymbol{x}_{0,p}^\top (\boldsymbol{X}_p^\top \boldsymbol{X}_p + n\lambda \boldsymbol{I})^{-1} \boldsymbol{X}_p^\top \boldsymbol{X}\beta}{\sigma \|\boldsymbol{x}_{0,p}\|_{(\boldsymbol{X}_p^\top \boldsymbol{X}_p + n\lambda \boldsymbol{I})^{-1} \boldsymbol{X}_p^\top \boldsymbol{X}_p (\boldsymbol{X}_p^\top \boldsymbol{X}_p + n\lambda \boldsymbol{I})^{-1}}}\right)^2 + D\beta^\top \beta + \text{Tr}\left(\boldsymbol{X}^\top \boldsymbol{X}\right)\right]\right]}{\|\boldsymbol{x}_{0,p}\|_{(\boldsymbol{X}_p^\top \boldsymbol{X}_p + n\lambda \boldsymbol{I})^{-1} \boldsymbol{X}_p^\top \boldsymbol{X}_p (\boldsymbol{X}_p^\top \boldsymbol{X}_p + n\lambda \boldsymbol{I})^{-1}}} \, d\beta d\boldsymbol{X}.$$

In the $m = 1$ case, because $\boldsymbol{x}_0$ is a row of $\boldsymbol{X}$, we condition on $\boldsymbol{x}_0$ but not on the remaining rows of $\boldsymbol{X}$. Without loss of generality, let $\boldsymbol{x}_0$ be the first row of $\boldsymbol{X}$ which can be done since $\boldsymbol{x}_0$ is selected uniformly and the rows of $\boldsymbol{X}$ are independent and identically distributed. Let $\tilde{\boldsymbol{X}} \in \mathbb{R}^{(n-1) \times D}$ denote $\boldsymbol{X}$ with its first row omitted such that $\boldsymbol{X} = [\boldsymbol{x}_0; \tilde{\boldsymbol{X}}]$. Following the same approach as the preceding marginalization, we have that

$$f_{\boldsymbol{x}_{0,p}^\top \hat{\beta} \mid \boldsymbol{x}_0, \{m=1\}}(x) = \int_{\mathbb{R}^{(n-1) \times D}} \int_{\mathbb{R}^D} f_{\boldsymbol{x}_{0,p}^\top \hat{\beta} \mid \boldsymbol{X}, \boldsymbol{x}_0, \{m=1\}} f_\beta f_{\tilde{\boldsymbol{X}}} \, d\beta d\tilde{\boldsymbol{X}}$$

Thus,

$$f_{\boldsymbol{x}_{0,p}^\top \hat{\beta} \mid \boldsymbol{x}_0, \{m=1\}}(x) = \frac{D^{\frac{D}{2}}}{\sigma(2\pi)^{\frac{nD+1}{2}}}$$
$$\times \int_{\mathbb{R}^{(n-1) \times D}} \int_{\mathbb{R}^D} \frac{\exp\left[-\frac{1}{2}\left[\left(\frac{x - \boldsymbol{x}_{0,p}^\top (\boldsymbol{X}_p^\top \boldsymbol{X}_p + n\lambda \boldsymbol{I})^{-1} \boldsymbol{X}_p^\top \boldsymbol{X}\beta}{\sigma \|\boldsymbol{x}_{0,p}\|_{(\boldsymbol{X}_p^\top \boldsymbol{X}_p + n\lambda \boldsymbol{I})^{-1} \boldsymbol{X}_p^\top \boldsymbol{X}_p (\boldsymbol{X}_p^\top \boldsymbol{X}_p + n\lambda \boldsymbol{I})^{-1}}}\right)^2 + D\beta^\top \beta + \text{Tr}\left(\tilde{\boldsymbol{X}}^\top \tilde{\boldsymbol{X}}\right)\right]\right]}{\|\boldsymbol{x}_{0,p}\|_{(\boldsymbol{X}_p^\top \boldsymbol{X}_p + n\lambda \boldsymbol{I})^{-1} \boldsymbol{X}_p^\top \boldsymbol{X}_p (\boldsymbol{X}_p^\top \boldsymbol{X}_p + n\lambda \boldsymbol{I})^{-1}}} \, d\beta d\tilde{\boldsymbol{X}}.$$

$\square$

# D   Theoretical Results for Regularized Linear Regression

**Theorem D.1.** *Membership advantage for Ridge-regularized linear regression. Consider the same setup as in Theorem 3.2 but now let $\hat{\beta}_\lambda = \left(\boldsymbol{X}_P^\top \boldsymbol{X}_P + n\lambda \boldsymbol{I}\right)^{-1} \boldsymbol{X}_P^\top$ for some $\lambda > 0$. Then, as $n, p, D \to \infty$ such that $\frac{p}{n} \to \gamma \in (1, \infty)$, we have:*

$$(\widehat{y_0} \mid m = 0, \boldsymbol{x}_0) \sim \mathcal{N}(0, \sigma_{0,\lambda}^2),$$
$$(\widehat{y_0} \mid m = 1, \boldsymbol{x}_0) \sim \mathcal{N}(0, \sigma_{1,\lambda}^2),$$

*where*

$$\sigma_{0,\lambda}^2 := \frac{g'(-\lambda)\gamma}{(1+g(-\lambda)\gamma)^2}\left(\sigma^2 + 1 - \frac{p}{D}\right)\frac{\|\boldsymbol{x}_{0,p}\|_2^2}{p} + (1 - 2\lambda g(-\lambda) + \lambda^2 g'(-\lambda))\frac{\|\boldsymbol{x}_{0,p}\|_2^2}{D}$$

$$\sigma_{1,\lambda}^2 := \left[\left(\frac{\lambda^2}{(\lambda + \gamma g(-\lambda))(\lambda + \gamma \frac{\|\boldsymbol{x}_{0,p}\|_2^2}{p}g(-\lambda))}\right)^2 \gamma g'(-\lambda)\frac{\|\boldsymbol{x}_{0,p}\|_2^2}{p}\right]\left(\sigma^2 + 1 - \frac{p}{D}\right)$$

$$+ \left(\frac{\gamma g(-\lambda)\frac{\|\boldsymbol{x}_{0,p}\|_2^2}{p}}{1 + \gamma g(-\lambda)\frac{\|\boldsymbol{x}_{0,p}\|_2^2}{p}}\right)^2 \left(\sigma^2 + \frac{\|\boldsymbol{x}_{0,\bar{p}}\|_2^2}{D}\right)$$

$$+ \left(1 - \frac{2\lambda g(-\lambda)}{1 + \frac{\gamma\|\boldsymbol{x}_{0,p}\|_2^2}{p}g(-\lambda)} + \frac{\lambda^2 g'(-\lambda)}{\left(1 + \frac{\gamma\|\boldsymbol{x}_{0,p}\|_2^2}{p}g(-\lambda)\right)^2}\right)\frac{\|\boldsymbol{x}_{0,p}\|_2^2}{D},$$

$$g(-\lambda) := \frac{-(1 - \gamma + \lambda) + \sqrt{(1 - \gamma + \lambda)^2 + 4\gamma\lambda}}{2\gamma\lambda}.$$

*Furthermore, in the case when $\sigma_{1,\lambda} > \sigma_{0,\lambda}$ and defining:*

$$\alpha_\lambda = \sqrt{\frac{\sigma_{0,\lambda}^2 \sigma_{1,\lambda}^2 \log\left(\frac{\sigma_{1,\lambda}^2}{\sigma_{0,\lambda}^2}\right)}{\sigma_{1,\lambda}^2 - \sigma_{0,\lambda}^2}},$$

*the optimal membership inference advantage is then:*

$$Adv(A_\lambda^*) = \mathbb{E}_{\boldsymbol{x}_0}\left[2\left\{\Phi\left(\frac{\alpha_\lambda}{\sigma_{0,\lambda}}\right) - \Phi\left(\frac{\alpha_\lambda}{\sigma_{1,\lambda}}\right)\right\}\right].$$

*Remark* D.2. The above result holds using the asymptotic distributions as $n, p, D \to \infty$. In Lemma C.3, we derive the non-asymptotic distributions for the predictions of the ridge-regularized least squares estimator, though they cannot be written in closed form.

*Proof of Theorem D.1* Let the input be $\boldsymbol{x}_{0,p} \in \mathbb{R}^p$. Similar to the proof of theorem 3.2, we can write

$$\boldsymbol{X}\beta + \epsilon = \boldsymbol{X}_p\beta_p + \boldsymbol{X}_{\bar{p}}\beta_{\bar{p}} + \epsilon = \boldsymbol{X}_p\beta_p + \eta$$

where $\eta = \boldsymbol{X}_{\bar{p}}\boldsymbol{\beta}_{\bar{p}} + \epsilon \sim \mathcal{N}\left(0, \left(1 + \sigma^2 - \frac{p}{D}\right)\boldsymbol{I}_n\right)$. Hence, we have

$$\widehat{y_0} = \boldsymbol{x}_{0,p}^\top\left(\boldsymbol{X}_p^\top\boldsymbol{X}_p + n\lambda\boldsymbol{I}_p\right)^{-1}\boldsymbol{X}_p^\top(\boldsymbol{X}_p\beta_p + \eta) = \boldsymbol{x}_{0,p}^\top\boldsymbol{X}_p^\top\left(\boldsymbol{X}_p\boldsymbol{X}_p^\top + n\lambda\boldsymbol{I}_p\right)^{-1}(\boldsymbol{X}_p\beta_p + \eta). \tag{7}$$

First note that in the case $m = 0$, we have

$$\widehat{y_0} = \boldsymbol{x}_{0,p}^\top\left(\boldsymbol{X}_p^\top\boldsymbol{X}_p + n\lambda\boldsymbol{I}_p\right)^{-1}\boldsymbol{X}_p^\top(\boldsymbol{X}_p\beta_p + \eta)$$

$$= \boldsymbol{x}_{0,p}^\top\left(\boldsymbol{X}_p^\top\boldsymbol{X}_p + n\lambda\boldsymbol{I}_p\right)^{-1}\boldsymbol{X}_p^\top\boldsymbol{X}_p\beta_p + \boldsymbol{x}_{0,p}^\top\left(\boldsymbol{X}_p^\top\boldsymbol{X}_p + n\lambda\boldsymbol{I}_p\right)^{-1}\boldsymbol{X}_p^\top\eta. \tag{8}$$

Letting the sample covariance matrix $\widehat{\boldsymbol{\Sigma}} \triangleq (1/n)\boldsymbol{X}_p^\top\boldsymbol{X}_p$, the first term in (8) can be written as

$$\boldsymbol{x}_{0,p}^\top\left(\boldsymbol{X}_p^\top\boldsymbol{X}_p + n\lambda\boldsymbol{I}_p\right)^{-1}\boldsymbol{X}_p^\top\boldsymbol{X}_p\beta_p = \boldsymbol{x}_{0,p}^\top\left(\widehat{\boldsymbol{\Sigma}} + \lambda\boldsymbol{I}_p\right)^{-1}\widehat{\boldsymbol{\Sigma}}\beta_p$$

$$= \boldsymbol{x}_{0,p}^\top\left(\widehat{\boldsymbol{\Sigma}} + \lambda\boldsymbol{I}_p\right)^{-1}\left(\widehat{\boldsymbol{\Sigma}} + \lambda\boldsymbol{I}_p - \lambda\boldsymbol{I}_p\right)\beta_p$$

$$= \left(\boldsymbol{x}_{0,p} - \lambda(\widehat{\boldsymbol{\Sigma}} + \lambda\boldsymbol{I}_p)^{-1}\boldsymbol{x}_{0,p}\right)^\top\beta_p.$$

Since $\beta_p \sim (0, \frac{1}{D}\boldsymbol{I}_D)$ using Lemma B.2, this converges to a gaussian with zero mean and variance

$$\lim_{n\to\infty}\frac{1}{D}\left\|\boldsymbol{x}_{0,p}-\lambda(\widehat{\boldsymbol{\Sigma}}+\lambda\boldsymbol{I}_p)^{-1}\boldsymbol{x}_{0,p}\right\|_2^2 = \lim_{n\to\infty}\frac{1}{D}\Big\{\|\boldsymbol{x}_{0,p}\|_2^2-2\lambda[\boldsymbol{x}_{0,p}^\top(\widehat{\boldsymbol{\Sigma}}+\lambda\boldsymbol{I}_p)^{-1}\boldsymbol{x}_{0,p}]$$
$$+\lambda^2[\boldsymbol{x}_{0,p}^\top(\widehat{\boldsymbol{\Sigma}}+\lambda\boldsymbol{I}_p)^{-2}\boldsymbol{x}_{0,p}]\Big\}$$
$$=\frac{\|\boldsymbol{x}_{0,p}\|_2^2}{D}(1-2\lambda g(-\lambda)+\lambda^2 g'(-\lambda))$$

where for the second equality, we have used the fact that using Lemma B.1 by setting $\boldsymbol{C}_n = (1/n)\boldsymbol{x}_{0,p}\boldsymbol{x}_{0,p}^\top$, as $n\to\infty$, almost surely,

$$\frac{1}{n}\boldsymbol{x}_{0,p}^\top(\widehat{\boldsymbol{\Sigma}}+\lambda\boldsymbol{I}_p)^{-1}\boldsymbol{x}_{0,p}\to\frac{1}{n}\|\boldsymbol{x}_{0,p}\|_2^2 g(-\lambda),\qquad\frac{1}{n}\boldsymbol{x}_{0,p}^\top(\widehat{\boldsymbol{\Sigma}}+\lambda\boldsymbol{I}_p)^{-2}\boldsymbol{x}_{0,p}\to\frac{1}{n}\|\boldsymbol{x}_{0,p}\|_2^2 g'(-\lambda).$$

For the second term in (8), using the rotationally invariance of gaussian distribution, without loss of generality, we can let $\eta$ to be $\boldsymbol{e}_1\|\eta\|_2$, where $\boldsymbol{e}_1$ is the first standard unit vector. Now, note that we have

$$\|\eta\|_2\boldsymbol{x}_{0,p}^\top\left(\boldsymbol{X}_p^\top\boldsymbol{X}_p+n\lambda\boldsymbol{I}_p\right)^{-1}\boldsymbol{X}_p^\top\boldsymbol{e}_1 = \|\eta\|_2\boldsymbol{x}_{0,p}^\top\left(\boldsymbol{x}_{1,p}\boldsymbol{x}_{1,p}^\top+\lambda n\boldsymbol{I}_p+\sum_{i=2}^n\boldsymbol{x}_{i,p}\boldsymbol{x}_{i,p}^\top\right)^{-1}\boldsymbol{x}_{1,p}$$

where $\boldsymbol{x}_{i,p}^\top\in\mathbb{R}^p$ is the $i$'th row of $\boldsymbol{X}_p$. Letting $\boldsymbol{A}_\lambda\triangleq\lambda\boldsymbol{I}_p+\frac{1}{n}\sum_{i=2}^n\boldsymbol{x}_{i,p}\boldsymbol{x}_{i,p}^\top$, by using the Sherman-Morrison formula, we have

$$\|\eta\|_2\boldsymbol{x}_{0,p}^\top\left(\boldsymbol{X}_p^\top\boldsymbol{X}_p+n\lambda\boldsymbol{I}_p\right)^{-1}\boldsymbol{X}_p^\top\boldsymbol{e}_1 = \|\eta\|_2\boldsymbol{x}_{0,p}^\top\left(\boldsymbol{x}_{1,p}\boldsymbol{x}_{1,p}^\top+n\boldsymbol{A}_\lambda\right)^{-1}\boldsymbol{x}_{1,p}$$
$$=\frac{\|\eta\|_2}{n}\boldsymbol{x}_{0,p}^\top\left(\boldsymbol{A}_\lambda^{-1}-\frac{\boldsymbol{A}_\lambda^{-1}\boldsymbol{x}_{1,p}\boldsymbol{x}_{1,p}^\top\boldsymbol{A}_\lambda^{-1}}{n+\boldsymbol{x}_{1,p}^\top\boldsymbol{A}_\lambda^{-1}\boldsymbol{x}_{1,p}}\right)\boldsymbol{x}_{1,p}$$
$$=\frac{\|\eta\|_2}{n}\boldsymbol{x}_{0,p}^\top\boldsymbol{A}_\lambda^{-1}\boldsymbol{x}_{1,p}\left(1-\frac{\boldsymbol{x}_{1,p}^\top\boldsymbol{A}_\lambda^{-1}\boldsymbol{x}_{1,p}}{n+\boldsymbol{x}_{1,p}^\top\boldsymbol{A}_\lambda^{-1}\boldsymbol{x}_{1,p}}\right)$$
$$=\|\eta\|_2\frac{\boldsymbol{x}_{0,p}^\top\boldsymbol{A}_\lambda^{-1}\boldsymbol{x}_{1,p}}{n+\boldsymbol{x}_{1,p}^\top\boldsymbol{A}_\lambda^{-1}\boldsymbol{x}_{1,p}}.$$

Note that using Lemma B.1 by setting $\boldsymbol{C}_n = (1/n)\boldsymbol{x}_{1,p}\boldsymbol{x}_{1,p}^\top$ and $\boldsymbol{C}_n = (1/n)\boldsymbol{x}_{0,p}\boldsymbol{x}_{0,p}^\top$, respectively, for $n,p\to\infty$, almost surely,

$$\frac{1}{n}\boldsymbol{x}_{1,p}^\top\boldsymbol{A}_\lambda^{-1}\boldsymbol{x}_{1,p}\to\gamma g(-\lambda),\qquad\frac{1}{n}\boldsymbol{x}_{0,p}^\top\boldsymbol{A}_\lambda^{-2}\boldsymbol{x}_{0,p}\to\frac{\|\boldsymbol{x}_{0,p}\|_2^2}{n}g'(-\lambda).$$

Thus, since $\boldsymbol{x}_{1,p}\sim(0,\boldsymbol{I}_p)$, using Lemma B.2, $\|\eta\|_2\boldsymbol{x}_{0,p}^\top\left(\boldsymbol{X}_p^\top\boldsymbol{X}_p+n\lambda\boldsymbol{I}_p\right)^{-1}\boldsymbol{X}_p^\top\boldsymbol{e}_1$ converges to a gaussian with mean zero and variance

$$\lim_{n\to\infty}\frac{\|\eta\|_2^2}{n^2(1+\gamma g(-\lambda))^2}\|\boldsymbol{A}_\lambda^{-1}\boldsymbol{x}_{0,p}\|_2^2 = \frac{\|\eta\|_2^2\|\boldsymbol{x}_{0,p}\|_2^2}{n^2(1+\gamma g(-\lambda))^2}g'(-\lambda)$$
$$=\frac{\|\boldsymbol{x}_{0,p}\|_2^2}{p}\frac{g'(-\lambda)\gamma}{1+\gamma g(-\lambda))^2}\left(\sigma^2+1-\frac{p}{D}\right).$$

Hence, by independence of $\beta_p$ and $\eta$, for $m=0$, as $n\to\infty$, such that $p/n=\gamma$, the output $\widehat{y}_0$ as in (7), converges in distribution to a gaussian with mean zero and variance

$$\frac{g'(-\lambda)\gamma}{(1+g(-\lambda)\gamma)^2}\left(\sigma^2+1-\frac{p}{D}\right)\frac{\|\boldsymbol{x}_{0,p}\|_2^2}{p}+(1-2\lambda g(-\lambda)+\lambda^2 g'(-\lambda))\frac{\|\boldsymbol{x}_{0,p}\|_2^2}{D}.$$

Now consider the $m=1$ case where the input belongs to training data. Without loss of generality, assume that the input is the first row of $\boldsymbol{X}_p$, i.e. $\boldsymbol{x}_0:=\boldsymbol{x}_1$. Note that in this case for $\eta=\boldsymbol{X}_{\overline{p}}\beta_{\overline{p}}+\epsilon$, we

have $\eta_i \sim \mathcal{N}\left(0, \left(\sigma^2 + \frac{\|\boldsymbol{x}_{1,\bar{p}}\|_2^2}{D}\right)\boldsymbol{I}_n\right)$, for $i = 1$, $\eta_i \sim \mathcal{N}\left(0, \left(1 + \sigma^2 - \frac{p}{D}\right)\right)$, for $i = 2, 3, \cdots, n$, and $\eta_i$'s are independent. We have

$$
\begin{aligned}
\widehat{y}_0 &= \boldsymbol{x}_{1,p}^\top \left(\boldsymbol{X}_p^\top \boldsymbol{X}_p + n\lambda \boldsymbol{I}_p\right)^{-1} \boldsymbol{X}_p^\top (\boldsymbol{X}_p \beta_p + \eta) \\
&= \boldsymbol{x}_{1,p}^\top \left(\boldsymbol{X}_p^\top \boldsymbol{X}_p + n\lambda \boldsymbol{I}_p\right)^{-1} \boldsymbol{X}_p^\top \boldsymbol{X}_p \beta_p + \boldsymbol{x}_{1,p}^\top \left(\boldsymbol{X}_p^\top \boldsymbol{X}_p + n\lambda \boldsymbol{I}_p\right)^{-1} \boldsymbol{X}_p^\top \eta \\
&= \boldsymbol{x}_{1,p}^\top \left(\widehat{\boldsymbol{\Sigma}} + \lambda \boldsymbol{I}_p\right)^{-1} \widehat{\boldsymbol{\Sigma}} \beta_p + \boldsymbol{x}_{1,p}^\top \left(\boldsymbol{X}_p^\top \boldsymbol{X}_p + n\lambda \boldsymbol{I}_p\right)^{-1} \boldsymbol{X}_p^\top \eta.
\end{aligned} \tag{9}
$$

The first term in (9) can be written as

$$
\begin{aligned}
\boldsymbol{x}_{1,p}^\top \left(\widehat{\boldsymbol{\Sigma}} + \lambda \boldsymbol{I}_p\right)^{-1} \widehat{\boldsymbol{\Sigma}} \beta_p &= \boldsymbol{x}_{1,p}^\top \beta_p - \lambda \boldsymbol{x}_{1,p}^\top \left(\widehat{\boldsymbol{\Sigma}} + \lambda \boldsymbol{I}_p\right)^{-1} \beta_p \\
&= \boldsymbol{x}_{1,p}^\top \beta_p - \lambda \boldsymbol{x}_{1,p}^\top \left[\frac{1}{n} \boldsymbol{x}_{1,p} \boldsymbol{x}_{1,p}^\top + \boldsymbol{A}_\lambda\right]^{-1} \beta_p \\
&= \boldsymbol{x}_{1,p}^\top \beta_p - \lambda \boldsymbol{x}_{1,p}^\top \left[\boldsymbol{A}_\lambda^{-1} - \frac{\boldsymbol{A}_\lambda^{-1} \boldsymbol{x}_{1,p} \boldsymbol{x}_{1,p}^\top \boldsymbol{A}_\lambda^{-1}}{n + \boldsymbol{x}_{1,p}^\top \boldsymbol{A}_\lambda^{-1} \boldsymbol{x}_{1,p}}\right] \beta_p \\
&= \boldsymbol{x}_{1,p}^\top \left(\boldsymbol{I}_p - \lambda \boldsymbol{A}_\lambda^{-1}\right) \beta_p + \frac{\lambda \boldsymbol{x}_{1,p}^\top \boldsymbol{A}_\lambda^{-1} \boldsymbol{x}_{1,p} \boldsymbol{x}_{1,p}^\top \boldsymbol{A}_\lambda^{-1} \beta_p}{n + \boldsymbol{x}_{1,p}^\top \boldsymbol{A}_\lambda^{-1} \boldsymbol{x}_{1,p}} \\
&= \boldsymbol{x}_{1,p}^\top \left(\boldsymbol{I}_p - \frac{\lambda}{1 + (1/n)\boldsymbol{x}_{1,p}^\top \boldsymbol{A}_\lambda^{-1} \boldsymbol{x}_{1,p}} \boldsymbol{A}_\lambda^{-1}\right) \beta_p
\end{aligned}
$$

Hence, since $\beta_i \sim (0, \frac{1}{D}\boldsymbol{I}_D)$, using Lemma B.2, The first term in (9) converges to a gaussian with zero mean and variance

$$
\begin{aligned}
\lim_{n\to\infty} \frac{1}{D} \left\|\widehat{\boldsymbol{\Sigma}}\left(\widehat{\boldsymbol{\Sigma}} + \lambda \boldsymbol{I}_p\right)^{-1} \boldsymbol{x}_{1,p}\right\|_2^2 &= \lim_{n\to\infty} \frac{1}{D} \left\|\left(\boldsymbol{I}_p - \frac{\lambda}{1 + (1/n)\boldsymbol{x}_{1,p}^\top \boldsymbol{A}_\lambda^{-1} \boldsymbol{x}_{1,p}} \boldsymbol{A}_\lambda^{-1}\right) \boldsymbol{x}_{1,p}\right\|_2^2 \\
&= \frac{\|\boldsymbol{x}_{1,p}\|_2^2}{D}\left(1 - \frac{2\lambda g(-\lambda)}{1 + \frac{\gamma\|\boldsymbol{x}_{1,p}\|_2^2}{p}g(-\lambda)} + \frac{\lambda^2 g'(-\lambda)}{\left(1 + \frac{\gamma\|\boldsymbol{x}_{1,p}\|_2^2}{p}g(-\lambda)\right)^2}\right).
\end{aligned}
$$

where for the second equality we have used the fact that using Lemma B.1 by setting $\boldsymbol{C}_n = (1/n)\boldsymbol{x}_{1,p}\boldsymbol{x}_{1,p}^\top$, as $n \to \infty$, such that $p/n = \gamma$, almost surely,

$$
\frac{1}{n}\boldsymbol{x}_{1,p}^\top \boldsymbol{A}_\lambda^{-1} \boldsymbol{x}_{1,p} \to \frac{1}{n}\|\boldsymbol{x}_{1,p}\|_2^2 g(-\lambda), \qquad \frac{1}{n}\boldsymbol{x}_{1,p}^\top \boldsymbol{A}_\lambda^{-2} \boldsymbol{x}_{1,p} \to \frac{1}{n}\|\boldsymbol{x}_{1,p}\|_2^2 g'(-\lambda).
$$

Now consider the second term in (9). It can be written as

$$
\begin{aligned}
\boldsymbol{x}_{1,p}^\top \left(\boldsymbol{X}_p^\top \boldsymbol{X}_p + n\lambda \boldsymbol{I}_p\right)^{-1} \boldsymbol{X}_p^\top \eta &= \boldsymbol{x}_{1,p}^\top \left(\boldsymbol{X}_p^\top \boldsymbol{X}_p + n\lambda \boldsymbol{I}_p\right)^{-1} \boldsymbol{x}_{1,p}^\top \eta_1 \\
&\quad + \sum_{i=2}^n \boldsymbol{x}_{1,p}^\top \left(\boldsymbol{X}_p^\top \boldsymbol{X}_p + n\lambda \boldsymbol{I}_p\right)^{-1} \boldsymbol{x}_{i,p}^\top \eta_i \\
&= A\eta_1 + \sum_{i=2}^n B_i \eta_i.
\end{aligned}
$$

First consider

$$A = \frac{1}{n}\boldsymbol{x}_{1,p}^\top \left( \frac{1}{n}\boldsymbol{x}_{1,p}\boldsymbol{x}_{1,p}^\top + \lambda\boldsymbol{I}_p + \underbrace{\frac{1}{n}\sum_{i=2}^{n}\boldsymbol{x}_{i,p}\boldsymbol{x}_{i,p}^\top}_{\boldsymbol{A}_\lambda} \right)^{-1} \boldsymbol{x}_{i,p}$$

$$= \frac{1}{n}\boldsymbol{x}_{1,p}^\top \left( \boldsymbol{A}_\lambda^{-1} - \frac{\boldsymbol{A}_\lambda^{-1}\boldsymbol{x}_{1,p}\boldsymbol{x}_{1,p}^\top\boldsymbol{A}_\lambda^{-1}}{n + \boldsymbol{x}_{1,p}^\top\boldsymbol{A}_\lambda^{-1}\boldsymbol{x}_{1,p}} \right) \boldsymbol{x}_{1,p}$$

$$= \frac{1}{n}\boldsymbol{x}_{1,p}^\top\boldsymbol{A}_\lambda^{-1}\boldsymbol{x}_{1,p} \left( 1 - \frac{\boldsymbol{x}_{1,p}^\top\boldsymbol{A}_\lambda^{-1}\boldsymbol{x}_{1,p}}{n + \boldsymbol{x}_{1,p}^\top\boldsymbol{A}_\lambda^{-1}\boldsymbol{x}_{1,p}} \right)$$

$$= \frac{\boldsymbol{x}_{1,p}^\top\boldsymbol{A}_\lambda^{-1}\boldsymbol{x}_{1,p}}{n + \boldsymbol{x}_{1,p}^\top\boldsymbol{A}_\lambda^{-1}\boldsymbol{x}_{1,p}} \xrightarrow{a.s.} \frac{\gamma g(-\lambda)(\|\boldsymbol{x}_{1,p}\|_2^2 /p)}{1 + \gamma g(-\lambda)(\|\boldsymbol{x}_{1,p}\|_2^2 /p)}.$$

Now, consider

$$B_2 = \frac{1}{n}\boldsymbol{x}_{1,p}^\top \left( \frac{1}{n}\boldsymbol{U}\boldsymbol{U}^\top + \lambda\boldsymbol{I}_p + \underbrace{\frac{1}{n}\sum_{i=3}^{n}\boldsymbol{x}_{i,p}\boldsymbol{x}_{i,p}^\top}_{\boldsymbol{A}_{2,\lambda}} \right)^{-1} \boldsymbol{x}_{i,p}; \quad \boldsymbol{U} \triangleq [\boldsymbol{x}_{1,p}\ \boldsymbol{x}_{2,p}]$$

$$= \frac{1}{n}\boldsymbol{x}_{1,p}^\top \left[ \boldsymbol{A}_{2,\lambda}^{-1} - \underbrace{\boldsymbol{A}_{2,\lambda}^{-1}\boldsymbol{U}\left( n\lambda\boldsymbol{I}_p + \boldsymbol{U}^\top\boldsymbol{A}_{2,\lambda}^{-1}\boldsymbol{U} \right)^{-1}\boldsymbol{U}^\top\boldsymbol{A}_{2,\lambda}^{-1}}_{\boldsymbol{C}_2} \right] \boldsymbol{x}_{2,p}.$$

We have

$$\boldsymbol{C}_2 = [\boldsymbol{A}_{2,\lambda}^{-1}\boldsymbol{x}_{1,p}\ \boldsymbol{A}_{2,\lambda}^{-1}\boldsymbol{x}_{2,p}] \begin{bmatrix} n\lambda + \boldsymbol{x}_{1,p}^\top\boldsymbol{A}_{2,\lambda}^{-1}\boldsymbol{x}_{1,p} & \boldsymbol{x}_{1,p}^\top\boldsymbol{A}_{2,\lambda}^{-1}\boldsymbol{x}_{2,p} \\ \boldsymbol{x}2,p^\top\boldsymbol{A}_{2,\lambda}^{-1}\boldsymbol{x}_{1,p} & n\lambda + \boldsymbol{x}_{2,p}^\top\boldsymbol{A}_{2,\lambda}^{-1}\boldsymbol{x}_{2,p} \end{bmatrix}^{-1} \begin{bmatrix} \boldsymbol{x}_{1,p}^\top\boldsymbol{A}_{2,\lambda}^{-1} \\ \boldsymbol{x}_{2,p}^\top\boldsymbol{A}_{2,\lambda}^{-1} \end{bmatrix}$$

$$= \left[ \underbrace{\left( n\lambda + \boldsymbol{x}_{1,p}^\top\boldsymbol{A}_{2,\lambda}^{-1}\boldsymbol{x}_{1,p} \right)\left( n\lambda + \boldsymbol{x}_{2,p}^\top\boldsymbol{A}_{2,\lambda}^{-1}\boldsymbol{x}_{2,p} \right) - \boldsymbol{x}_{1,p}^\top\boldsymbol{A}_{2,\lambda}^{-1}\boldsymbol{x}_{2,p}\boldsymbol{x}2,p^\top\boldsymbol{A}_{2,\lambda}^{-1}\boldsymbol{x}_{1,p}}_{D_2} \right]^{-1} \widetilde{\boldsymbol{C}}_2.$$

We have

$$\widetilde{\boldsymbol{C}}_2 = [\boldsymbol{A}_{2,\lambda}^{-1}\boldsymbol{x}_{1,p}\ \boldsymbol{A}_{2,\lambda}^{-1}\boldsymbol{x}_{2,p}] \begin{bmatrix} n\lambda + \boldsymbol{x}_{2,p}^\top\boldsymbol{A}_{2,\lambda}^{-1}\boldsymbol{x}_{2,p} & -\boldsymbol{x}_{1,p}^\top\boldsymbol{A}_{2,\lambda}^{-1}\boldsymbol{x}_{2,p} \\ -\boldsymbol{x}_{2,p}^\top\boldsymbol{A}_{2,\lambda}^{-1}\boldsymbol{x}_{1,p} & n\lambda + \boldsymbol{x}_{1,p}^\top\boldsymbol{A}_{2,\lambda}^{-1}\boldsymbol{x}_{1,p} \end{bmatrix} \begin{bmatrix} \boldsymbol{x}_{1,p}^\top\boldsymbol{A}_{2,\lambda}^{-1} \\ \boldsymbol{x}_{2,p}^\top\boldsymbol{A}_{2,\lambda}^{-1} \end{bmatrix}$$

$$= \left( n\lambda + \boldsymbol{x}_{2,p}^\top\boldsymbol{A}_{2,\lambda}^{-1}\boldsymbol{x}_{2,p} \right)\boldsymbol{A}_{2,\lambda}^{-1}\boldsymbol{x}_{1,p}\boldsymbol{x}_{1,p}^\top\boldsymbol{A}_{2,\lambda}^{-1} - \boldsymbol{x}_{1,p}^\top\boldsymbol{A}_{2,\lambda}^{-1}\boldsymbol{x}_{2,p}\boldsymbol{A}_{2,\lambda}^{-1}\boldsymbol{x}_{1,p}\boldsymbol{x}_{2,p}^\top\boldsymbol{A}_{2,\lambda}^{-1}$$

$$- \boldsymbol{x}_{2,p}^\top\boldsymbol{A}_{2,\lambda}^{-1}\boldsymbol{x}_{1,p}\boldsymbol{A}_{2,\lambda}^{-1}\boldsymbol{x}_{2,p}\boldsymbol{x}_{1,p}^\top\boldsymbol{A}_{2,\lambda}^{-1} + \left( n\lambda + \boldsymbol{x}_{1,p}^\top\boldsymbol{A}_{2,\lambda}^{-1}\boldsymbol{x}_{1,p} \right)\boldsymbol{A}_{2,\lambda}^{-1}\boldsymbol{x}_{2,p}\boldsymbol{x}_{2,p}^\top\boldsymbol{A}_{2,\lambda}^{-1}.$$

Thus,

$$B_2 = \frac{1}{n}\boldsymbol{x}_{1,p}^\top \left[ \boldsymbol{A}_{2,\lambda}^{-1} - \frac{\widetilde{\boldsymbol{C}}_2}{D_2} \right] \boldsymbol{x}_{2,p}$$

$$= \frac{\boldsymbol{x}_{1,p}^\top}{n} \left\{ \boldsymbol{A}_{2,\lambda}^{-1} - \left[ \left( n\lambda + \boldsymbol{x}_{1,p}^\top\boldsymbol{A}_{2,\lambda}^{-1}\boldsymbol{x}_{1,p} \right)\left( n\lambda + \boldsymbol{x}_{2,p}^\top\boldsymbol{A}_{2,\lambda}^{-1}\boldsymbol{x}_{2,p} \right) - \boldsymbol{x}_{1,p}^\top\boldsymbol{A}_{2,\lambda}^{-1}\boldsymbol{x}_{2,p}\boldsymbol{x}_{2,p}^\top\boldsymbol{A}_{2,\lambda}^{-1}\boldsymbol{x}_{1,p} \right]^{-1} \right.$$

$$\left[ \left( n\lambda + \boldsymbol{x}_{2,p}^\top\boldsymbol{A}_{2,\lambda}^{-1}\boldsymbol{x}_{2,p} \right)\boldsymbol{A}_{2,\lambda}^{-1}\boldsymbol{x}_{1,p}\boldsymbol{x}_{1,p}^\top\boldsymbol{A}_{2,\lambda}^{-1} - \boldsymbol{x}_{1,p}^\top\boldsymbol{A}_{2,\lambda}^{-1}\boldsymbol{x}_{2,p}\boldsymbol{A}_{2,\lambda}^{-1}\boldsymbol{x}_{1,p}\boldsymbol{x}_{2,p}^\top\boldsymbol{A}_{2,\lambda}^{-1} \right.$$

$$\left. \left. - \boldsymbol{x}_{2,p}^\top\boldsymbol{A}_{2,\lambda}^{-1}\boldsymbol{x}_{1,p}\boldsymbol{A}_{2,\lambda}^{-1}\boldsymbol{x}_{2,p}\boldsymbol{x}_{1,p}^\top\boldsymbol{A}_{2,\lambda}^{-1} + \left( n\lambda + \boldsymbol{x}_{1,p}^\top\boldsymbol{A}_{2,\lambda}^{-1}\boldsymbol{x}_{1,p} \right)\boldsymbol{A}_{2,\lambda}^{-1}\boldsymbol{x}_{2,p}\boldsymbol{x}_{2,p}^\top\boldsymbol{A}_{2,\lambda}^{-1} \right] \right\}\boldsymbol{x}_{2,p}.$$

using Lemma B.1 by setting $\boldsymbol{C}_n = (1/n)\boldsymbol{x}\boldsymbol{x}^\top$, as $n, p \to \infty$, such that $p/n = \gamma$, almost surely,

$$\frac{1}{n}\boldsymbol{x}^\top \boldsymbol{A}_\lambda^{-2} \boldsymbol{x} \to \frac{\|\boldsymbol{x}\|_2^2}{n} g(-\lambda).$$

Hence, letting $n \to \infty$, $B_2$ converges weakly to

$$
\begin{aligned}
B_2' = {} & \frac{1}{n}\boldsymbol{x}_{1,p}^\top \boldsymbol{A}_{2,\lambda}^{-1} \Bigg\{ \boldsymbol{A}_{2,\lambda} - \left[ \left( \lambda + g(-\lambda)\frac{\|\boldsymbol{x}_{1,p}\|_2^2}{n} \right) \left( \lambda + g(-\lambda)\frac{\|\boldsymbol{x}_{2,p}\|_2^2}{n} \right) - \left( \frac{\boldsymbol{x}_{1,p}^\top \boldsymbol{A}_{2,\lambda}^{-1} \boldsymbol{x}_{2,p}}{n} \right)^2 \right]^{-1} \\
& \left[ \left( \lambda + g(-\lambda)\frac{\|\boldsymbol{x}_{2,p}\|_2^2}{n} \right) \frac{\boldsymbol{x}_{1,p}\boldsymbol{x}_{1,p}^\top}{n} - \left( \frac{\boldsymbol{x}_{1,p}^\top \boldsymbol{A}_{2,\lambda}^{-1}\boldsymbol{x}_{2,p}}{n} \right) \left( \frac{\boldsymbol{x}_{1,p}\boldsymbol{x}_{2,p}^\top}{n} + \frac{\boldsymbol{x}_{2,p}\boldsymbol{x}_{1,p}^\top}{n} \right) \right. \\
& \left. + \left( \lambda + g(-\lambda)\frac{\|\boldsymbol{x}_{1,p}\|_2^2}{n} \right) \frac{\boldsymbol{x}_{2,p}\boldsymbol{x}_{2,p}^\top}{n} \right] \Bigg\} \boldsymbol{A}_{2,\lambda}^{-1}\boldsymbol{x}_{2,p} \\
= {} & \frac{1}{n}\boldsymbol{x}_{1,p}^\top \boldsymbol{A}_{2,\lambda}^{-1}\boldsymbol{x}_{2,p} \frac{\lambda^2}{\left( \lambda + g(-\lambda)\frac{\|\boldsymbol{x}_{1,p}\|_2^2}{n} \right) \left( \lambda + g(-\lambda)\frac{\|\boldsymbol{x}_{2,p}\|_2^2}{n} \right) - \left( \frac{\boldsymbol{x}_{1,p}^\top \boldsymbol{A}_{2,\lambda}^{-1}\boldsymbol{x}_{2,p}}{n} \right)^2}.
\end{aligned}
$$

Note that by LLN, $(1/n)\|\boldsymbol{x}_{2,p}\|_2^2 \to \gamma$ and $(1/n^2)\left( \boldsymbol{x}_{1,p}^\top \boldsymbol{A}_{2,\lambda}^{-1}\boldsymbol{x}_{2,p} \right)^2 \to 0$. Further, since $\boldsymbol{x}_{2,p} \sim (0, \boldsymbol{I}_p)$, using Lemma B.2, as $n \to \infty$, $(1/\sqrt{n})\boldsymbol{x}_{1,p}^\top \boldsymbol{A}_{2,\lambda}^{-1}\boldsymbol{x}_{2,p}$ converges to a gaussian with mean zero and variance

$$\lim_{n\to\infty} \frac{1}{n}\|\boldsymbol{A}_{2,\lambda}^{-1}\boldsymbol{x}_{1,p}\|_2^2 = \lim_{n\to\infty} \frac{1}{n}\boldsymbol{x}_{1,p}^\top \boldsymbol{A}_{2,\lambda}^{-2}\boldsymbol{x}_{1,p} = \frac{1}{n}\|\boldsymbol{x}_{1,p}\|_2^2 g'(-\lambda)$$

where for the second equality we have used Lemma B.1 with $\boldsymbol{C}_n = (1/n)\boldsymbol{x}_{1,p}\boldsymbol{x}_{1,p}^\top$. Hence, $B_2$ converges weakly to

$$\frac{\widetilde{\sigma}}{n}\boldsymbol{x}_{1,p}^\top \boldsymbol{A}_{2,\lambda}^{-1}\boldsymbol{x}_{2,p} \xrightarrow{\mathcal{D}} \mathcal{N}\left( 0, \frac{\gamma\widetilde{\sigma}^2 \|\boldsymbol{x}_{1,p}\|_2^2}{pn} g'(-\lambda) \right)$$

where

$$\widetilde{\sigma} = \frac{\lambda^2}{\lambda^2 + \gamma\lambda g(-\lambda)\left[ 1 + \frac{\|\boldsymbol{x}_{1,p}\|_2^2}{p} \right] + \gamma^2 m^2(-\lambda)\frac{\|\boldsymbol{x}_{1,p}\|_2^2}{p}}.$$

By symmetry over $i$, $\sum_{i=2}^n B_i^2 = (n-1)B_2^2$ that converges almost surely to $\frac{\gamma\widetilde{\sigma}^2 \|\boldsymbol{x}_{1,p}\|_2^2}{p} g'(-\lambda)$. Therefore using Lemma B.2

$$\sum_{i=2}^n B_i \eta_i \xrightarrow{\mathcal{D}} \mathcal{N}\left( 0, \left( \sigma^2 + 1 - \frac{p}{D} \right) \gamma\tilde{\sigma}^2 \frac{\|\boldsymbol{x}_{1,p}\|_2^2 g'(-\lambda)}{p} \right).$$

Thus, by independence of $\beta_p$ and $\eta_i$'s, as $n, p \to \infty$ the output $y$ converges in distribution to a gaussian with zero mean and variance

$$
\begin{aligned}
& \left[ \left( \frac{\lambda^2}{(\lambda + \gamma g(-\lambda))(\lambda + \gamma\frac{\|\boldsymbol{x}_{1,p}\|_2^2}{p} g(-\lambda))} \right)^2 \gamma g'(-\lambda)\frac{\|\boldsymbol{x}_{1,p}\|_2^2}{p} \right] \left( \sigma^2 + 1 - \frac{p}{D} \right) \\
& + \left( \frac{\gamma g(-\lambda)\frac{\|\boldsymbol{x}_{1,p}\|_2^2}{p}}{1 + \gamma g(-\lambda)\frac{\|\boldsymbol{x}_{1,p}\|_2^2}{p}} \right)^2 \left( \sigma^2 + \frac{\|\boldsymbol{x}_{1,\bar{p}}\|_2^2}{D} \right) \\
& + \left( 1 - \frac{2\lambda g(-\lambda)}{1 + \frac{\gamma\|\boldsymbol{x}_{1,p}\|_2^2}{p} g(-\lambda)} + \frac{\lambda^2 g'(-\lambda)}{\left( 1 + \frac{\gamma\|\boldsymbol{x}_{1,p}\|_2^2}{p} g(-\lambda) \right)^2} \right) \frac{\|\boldsymbol{x}_{1,p}\|_2^2}{D},
\end{aligned}
$$

which completes the proof. $\qquad\square$

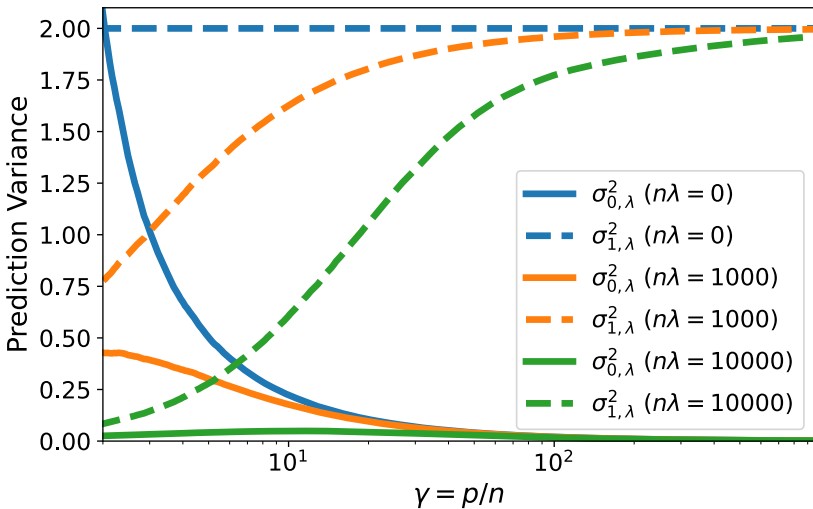

Figure 5: Theoretical variances of the predictions $\widehat{y}_0$ by ridge regularized linear regression models for the Gaussian data setting with $n = 10^3$, $D = 10^7$, and $\sigma = 1$ on a single sampled $\boldsymbol{x}_0$ for when $\boldsymbol{x}_0$ is a test point ($\sigma^2_{0,\lambda}$) and when $\boldsymbol{x}_0$ is a training point ($\sigma^2_{1,\lambda}$) for different amounts of regularization $\lambda$. While increased ridge regularization decreases the variance $\sigma^2_{0,\lambda}$ on training point predictions, it also decreases the variance $\sigma^2_{1,\lambda}$ for test points in such a way that the two distributions become easier to distinguish. As such, membership inference is easier for ridge regularized models in this setting.

## E  Experimental Implementation Details

All experiments were ran only on CPUs on our internal servers without GPU processing. Processors used may have included Intel Xeon CPU E5-2630 (256GB RAM), Intel Xeon Silver 4214 CPU (192GB RAM), Intel Xeon Platinum 8260 CPU (192GB RAM), and AMD Ryzen Threadripper 1900X (32GB RAM). Our code is primarily written in Python and mainly uses numpy implementations of linear algebra operations. Please refer to our code on the Github page for more details.

The histograms in Figure 1 are obtained as follows. We first sample a vector $\boldsymbol{x}_0 \sim \mathcal{N}(0, \mathbf{I}_D)$, where $D = 20,000$. Then, for each $p = \gamma n$, we perform the following procedure 20,000 times. We sample $\beta \sim \mathcal{N}(0, \frac{1}{D}\mathbf{I}_D)$. Then, we sample an $n \times D$ matrix $\mathbf{X}$ such that each element is iid standard normal. We then generate the ground truth vector $\boldsymbol{y} = \mathbf{X}\beta + \epsilon$, where $\epsilon$ is an $n$-dimensional vector whose elements are iid standard normal. We obtain the least squares estimates $\hat{\beta}$ on the first $p$ columns of $\mathbf{X}$ and on the vector $\mathbf{y}$ using numpy's lstsq function. Finally, we collect the $\widehat{y}_0 = \boldsymbol{x}_{0,p}^\top \hat{\beta}$ of all 20,000 models to form the blue histograms in Figure 1. The orange histograms are formed the same way except that the first row of $\mathbf{X}$ is replaced with $\boldsymbol{x}_0$ and the first element of $\boldsymbol{y}$ is replaced with $y_0 = \boldsymbol{x}_0^\top \beta + \epsilon_0$ for $\epsilon_0 \sim \mathcal{N}(0,1)$.

The experiment in Figure 2b is performed as follows. In the experiment, we estimate the optimal membership advantage. Since the optimal MI adversary requires knowledge of the linear regression model's output distributions when a data point $\boldsymbol{x}_0$ is in its training dataset ($m = 1$) and when $\boldsymbol{x}_0$ is not ($m = 0$), we approximate these distributions by forming discrete histograms. To obtain the samples for the histograms, we use the same procedure as detailed in the previous paragraph, except that we obtain 100,000 samples for each histogram for increased precision. From these samples, the discrete histograms for $(\widehat{y}_0 \mid m = 0)$ and $(\widehat{y}_0 \mid m = 1)$ for a given $\gamma$ are then formed by splitting the interval between the minimum and maximum values over both $(\widehat{y}_0 \mid m = 0)$ and $(\widehat{y}_0 \mid m = 1)$ into 150 equally spaced bins. The histograms are normalized so that they represent probability mass functions (i.e. the bin counts sum to 1). Finally, treating the two histograms as probability mass functions, the membership advantage is calculated according to Definition 2.1. For Figure 2b, this procedure is repeated 20 times, each with a newly sampled $\boldsymbol{x}_0$, and the mean membership advantage over the 20 experiments is plotted.

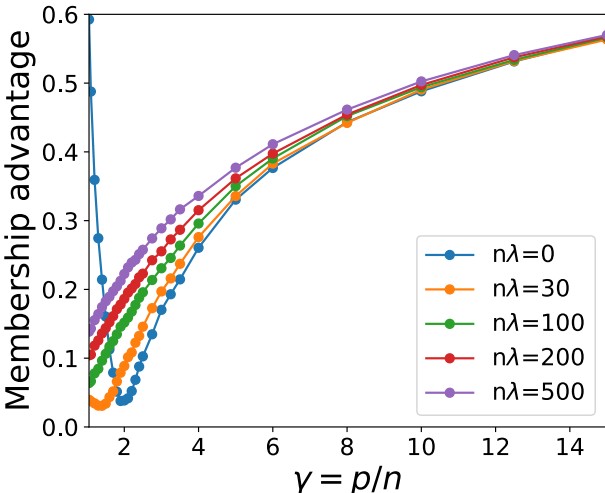

Figure 6: Experimental membership advantages for ridge-regularized linear regression on Gaussian data with $n = 100$, $D = 3000$, and $\sigma = 1$ for different regularization strengths $\lambda$. As predicted by our theory, membership advantage increases with additional regularization in the sufficiently overparameterized regime. This experiment verifies our theoretical findings.

The experiments in Figure 4 are also obtained by approximating the optimal MI adversary with discrete histograms as in the previous paragraph. The only difference is in how the datasets ($\boldsymbol{X}$, $\boldsymbol{y}$, etc.) are sampled. Specifically, they are sampled according to the distributions for each experiment detailed in Section 5. Again, the histograms are formed by splitting the model's prediction interval for each $\gamma$ into 150 equally spaced bins. 20 experiments are performed for each data model, with the means and standard errors reported in the figures.

## F  Experimental Verification of Ridge Theory

We verify our theoretical finding that ridge regression increases membership advantage on linear regression models with Gaussian data in the overparameterized regime. The experiment follows the procedure detailed in Section E for Figure 2b except that we only sample 50,000 datasets for each of $m = 0$ and $m = 1$ for each $\gamma$ and each $\lambda$ for computational efficiency. For this experiment, we set $n = 100$, $D = 3,000$, and $\sigma = 1$, as in Figure 2b. The results, shown in Figure 6, closely resemble the trend shown in the theoretical plot in Figure 3a, thus verifying our theory.