# OpenReview forum: "Parameters or Privacy: A Provable Tradeoff Between Overparameterization and Membership Inference"
_NeurIPS.cc/2022/Conference — NeurIPS 2022 Accept_

### Official Review · Reviewer_uY2r · 2022-07-10

**Rating:** 5
**Confidence:** 3
**Soundness:** 3 good
**Presentation:** 3 good
**Contribution:** 1 poor

**Summary:**

This paper first mathematically formalizes the existing empirical observation that larger models (linear regression) memorize more data (through the membership inference looking glass), and then provides some experiments on slightly more complicated models to further support the conclusion that having smaller models is better for privacy, even better than adding noise/ regularizing during training.

**Questions:**

1. As mentioned in the weaknesses, I am curious to see the distribution of MIA success on individual samples, when comparing noise addition to making model size smaller.

**Limitations:**

Apart from the questions above, the limitations are addressed.

**Strengths And Weaknesses:**

Strengths:

1. The observation that increasing the ridge regularization coefficient increases memorization is new and interesting.


Weaknesses:

1. The fact that larger models memorize more has been shown abundantly in the past (not theoretically, but this paper only looks at linear regression theoretically), for instance in [1] and [2] this is shown for GPT-2 and BERT families, respectively.

2. The conclusion that downsizing models is better than adding noise for privacy is too broad and even detrimental. I think it is important to clarify that smaller models only memorize less, 'on average'.  I wonder if the observation of figure 3c holds only when we look at average memorization. I would guess if we look at individual sample memorization, the worst case  would be better on the noise addition method (which is similar to DP).  I think calling smaller models a privacy mitigation/defense could have negative consequences.

3. This final point is not completely relevant, but as far as the broad conclusion of smaller models are better for privacy goes, there is another way in which it is not accurate. Recent work has shown that  when DP-SGD is used for training models, it is in fact better to have larger, pre-trained models that you fine-tune on [3,4]. I understand that this is outside the scope of this paper.

Summary: The paper formalizes empirical observations and existing heuristics, but other than that I do not see any added insights that would change the way we deploy or think about models. Yes smaller models are better in terms of memorization, when you have smaller data, but as datasets get larger we need to scale up. I will increase my score if my understanding of parts of the paper is lacking and I have misunderstood something.

[1] Carlini N, Tramer F, Wallace E, Jagielski M, Herbert-Voss A, Lee K, Roberts A, Brown T, Song D, Erlingsson U, Oprea A. Extracting training data from large language models. In30th USENIX Security Symposium (USENIX Security 21) 2021 (pp. 2633-2650).

[2] Mireshghallah F, Goyal K, Uniyal A, Berg-Kirkpatrick T, Shokri R. Quantifying Privacy Risks of Masked Language Models Using Membership Inference Attacks. arXiv e-prints. 2022 Mar:arXiv-2203.

[3] Li X, Tramer F, Liang P, Hashimoto T. Large language models can be strong differentially private learners. arXiv preprint arXiv:2110.05679. 2021 Oct 12. (ICLR)

[4] Yu D, Naik S, Backurs A, Gopi S, Inan HA, Kamath G, Kulkarni J, Lee YT, Manoel A, Wutschitz L, Yekhanin S. Differentially private fine-tuning of language models. arXiv preprint arXiv:2110.06500. 2021 Oct 13.

---

> ### Author Response · Authors · 2022-08-02
> **Response to Reviewer uY2r**
>
> We thank the reviewer for their insightful comments on our paper. They have provided important comments regarding our work that we respond to here.
>
> **How our work relates to existing work**
>
> The reviewer has correctly pointed out that some works ([1] and [2]) have empirical observations of membership inference (MI) attacks having higher accuracy on a larger model than a smaller model. Our main contribution is not in showing the existence of the phenomenon that increased parameterization leads to increased MI vulnerability. Rather, our main contribution is characterizing and explaining it precisely. As the reviewer points out, we are the first to analytically characterize this phenomenon and explain its source theoretically, and we believe this is an important step towards a stronger understanding. Further, our work analytically quantifies the phenomenon by specifying the asymptotic rates for MI vulnerability. This mathematical characterization provides much deeper insights into the tradeoff between parameterization and privacy that [1] and [2] do not.
>
> We also wish to emphasize our discovery that the parameterization-privacy trade-off is not caused by models memorizing more as they get larger. In the overparameterized regime that we study, all models, regardless of size, memorize equally: they all achieve exactly zero training error. We find that, even in this regime, increasing model size increases MI vulnerability, even if it has no effect on training data memorization (and even achieves lower test error). Our theory reveals that the root cause of this phenomenon is the decreased variance in the model’s predictions on test points and not increased memorization of training points. We believe this explanation of the parameterization-privacy trade-off to be an illuminating finding that is uncovered for the first time by our theory.
>
> **Average case vs. worst case guarantees for membership inference**
>
> We agree with the reviewer that our results focus on the average-case MI accuracy (as is typically studied for MI) and not on any worst-case metric (as is typically studied for differential privacy). We will clarify this in the revision and emphasize that we do not seek worst-case guarantees.
>
> We do note that, in the asymptotic regime, almost all samples $x_0$ will have the same membership advantage as the average case. Theorem 3.2 shows that the only term in the membership advantage dependent on $x_0$ are the variances, which depend on $||x_{0,p}||$ and $||x_0||$. In the asymptotic regime, as $p$ and $D$ grow large, these norms become fixed constants with probability 1 by the strong law of large numbers. Hence, the instance specific rate holds uniformly for all possible $x_0$ in this setting.
>
> Further, we repeated the experiment of Figure 3c where we instead sampled $x_0$ 10,000 times and recorded the worst-case membership advantage over the sampled $x_0$s. The results are similar to our paper's Figure 3c, with model downsizing having a better trade-off than noise addition. We can add these results (and distributions) in our revision appendix. We acknowledge this is not a formal guarantee, which would be an interesting avenue for future study.
>
> That being said, we do agree that we need to clarify that our findings are primarily focused on the average-case metric and may not reflect worst-case performance in the non-asymptotic regime.
>
> **On our observation that downsizing models might be good for privacy**
>
> We agree with the reviewer that making a broad conclusion that downsizing models is a universally effective defense for privacy could be both inaccurate and misleading, especially since it is not obvious whether our findings would hold for models employing modern privacy-preserving techniques such as DP-SGD. We do not intend to make such a claim, and we will more carefully clarify this in our paper. Indeed, our paper simply makes the observation that, for the data model we consider, tuning the model size improves model privacy.
>
> There have been many techniques proposed to increase model privacy (such as DP-SGD), and we agree that it is important to study how model size relates with them as well. We do hope that our observation of model downsizing’s effectiveness for MI protection (including vs. noise addition) with our data model spurs insights into how model size can play a role in designing defenses for privacy. More comprehensive future study is definitely required to prescribe a more general defense strategy.
>
> We will definitely clarify these points in our revision and ensure that it does not come across as preaching model downsizing to be a universally good defense for privacy.
>
> **On model size scaling up with dataset**
>
> The reviewer remarked that, as datasets grow larger, we would need to scale up our model. We agree and clarify that we use $\gamma = \frac{p}{n}$ as our measure of model size. As it is the ratio of the number of parameters to the dataset size, it does take dataset scaling into account.

---

### Official Review · Reviewer_YQ6y · 2022-07-11

**Rating:** 6
**Confidence:** 3
**Soundness:** 3 good
**Presentation:** 3 good
**Contribution:** 3 good

**Summary:**

This paper considers the problem of membership inference, i.e. guessing whether a given sample was in a model's training set. The authors propose to use the tools of random matrix theory in the asymptotic regime to analyze membership inference in the simple case of a linear model on Gaussian data. The authors start by deriving the explicit advantage of the attack in the asymptotic regime. Further derivations allow the author to analyze several interesting machine learning ingredients, starting from L2 regularization (ridge regression). There, the authors show that regularization has the counter intuitive effect of increasing the performance of membership inference attacks. The authors go on to show that feature reduction is more effective than noise addition in the generalization error - membership advantage tradeoff. Finally, the authors consider more complex models such as linear regression on latent space models or noise free time series regression.

**Questions:**

- Can you clarify if ||x_0, p||^2  ~ p in Theorem 3.2?

**Limitations:**

The authors have addressed limitations and potential negative societal impact (Appendix A.1 and A.2)

**Strengths And Weaknesses:**

Strengths
- The paper tackles an important problem
- The paper is well-written and easy to follow
- The setup of using random matrix theory to analyze membership inference on linear models is interesting and to the best of my knowledge new
- The results predicted by the theory seem to match closely the empirical results on Gaussian data

Weaknesses
- I am not very familiar with random matrix theory but some notations seem surprising to me, e.g. in Theorem 3.2 ||x_0, p||^2 should be p in the asymptotic regime if I understand correctly.
- I find the term "generalization error" to be ambiguous, in this case it refers really to the "test" performance rather than the train-test gap.

---

> ### Author Response · Authors · 2022-08-02
> **Response to Reviewer YQ6y**
>
> Thank you for taking the time to carefully review our work and for your suggestions for improvement.
>
> **On the norm of $||x_{0,p}||$**
>
> In Theorem 3.2, we take x_0 to be a fixed sample that we condition on. Hence, $||x_{0,p}||$ is a fixed value in this setting. You are correct though that this norm asymptotically equals $\Theta(p)$. We will clarify this point in the revision.
>
> **On the phrase “generalization error”**
>
> You are correct that we are focused on the performance on new test points rather than on the train-test gap. We opted for the phrase “generalization error” over “test error” since we are computing the exact expected error on the data distribution rather than the performance on a finite, held-out test set as we take the term “test error” to commonly mean. Regardless, since we are in the overparameterized setting, the training error is always 0, hence the train-test gap and test error coincide, and we are open to adjusting the terminology as long as it conveys that the error is not on a finite set but is with respect to the data distribution.

---

### Official Review · Reviewer_ggEs · 2022-07-13

**Rating:** 7
**Confidence:** 4
**Soundness:** 3 good
**Presentation:** 3 good
**Contribution:** 3 good

**Summary:**

Membership inference attacks are one way to break (differential) privacy by identifying whether a given instance belongs to the dataset of an accessible model trained on that data set or not. Neural nets (and sometimes other models) are shown to be susceptible to such attacks, due to their memorization.

This paper aims to show the intuitive fact that a learning algorithm with more parameters can provably and empirically be more susceptible to membership inference attacks.

The focus of the paper is on linear regression and it shows that learning such functions with more parameters (in a formal sense) not only leads to more accuracy but also more vulnerability to membership inference.  The exact formalization is by picking a challenge instance with probability ½ from the training set and ½ as a fresh instance and computing “advantage” eps such that adversary's chance of winning is $(1+\epsilon)/2$ (achieving advantage $\epsilon=0$ is trivial). To control the parameters the learner is limited to picking the model to be of the form $(x_1,...x_p,0,0,...)$ with total dimension D (and effective dimension p) when the data examples are of dimension D. The paper then analytically computes the exact optimal strategies for guessing the correct answer (with maximum probability) by using the Bayes optimal rule for hypothesis testing.

The paper then studies a few other scenarios:

-specific (ridge) regularization turns out to be not helpful for preventing MI.

-it is shown that for the same generalization error, using fewer parameters is more effective (against MI attacks) than adding *independent* noise to the answers. This should not be confused with adding noise in a central way to get DP schemes.

-The paper then studies the same phenomenon in other contexts such as time series and latent spaces. I particularly liked the time series setting, because it models an interesting scenario in which a person's dates of visiting the hospital might be deduced from the (approximate) Fourier representation of the sequence of visits.


**Questions:**

I don't have a questions that fundamentally changes my mind about the paper. Some minor questions are listed above.

**Limitations:**

I see no serious societal down side with this paper. Quite the contrary, it studies an important question about privacy and it does it formally. However I think the paper should address its limitations discussed above (as weaknesses) in more detail.


**Strengths And Weaknesses:**

The strength is that the paper formally studies a well motivated question and is among the first to do it *formally*. This is a big plus Understanding what causes membership inference is an important question and overparameterization is a well-known suspect. It is helpful to have a deeper understanding of this phenomenon.

On the down side:

-It is not clear from this work whether a method *designed* for resisting MI would be hurt in the overparameterized regime. In particular, …

-No *comparison* (between this paper’s methods for positive solutions resisting attacks) with differentially private methods is made. Ideally, I would have loved to see a comparison between the “pruning” method and DP-linear regression.

-The study is only for regression.

-One major limitation is that the membership inference’s adversary is very limited. The typical (I’d say standard) adversary shall be able to at least make more than 1 query to the model (or even have the model given in a full whilte box setting) while the attacker of this paper gets a *single* query. The paper seems to address this when saying “We emphasize that our results on the blind adversary lower bound the performance of adversaries…” which is true, but it is also true that by limiting the adversary the comparison baseline changes and you *compare* limited adversaries for different levels of parameterization.

-Overparameterization is a major issue for neural nets and NNs also suffer from MI, but the formal study here is for the clean setting of basically everything being Gaussian (and the tasks being linear regression and variants).

-The paper over-interprets its findings and ends with, in my opinion, dangerous recommendations such as “simply reducing the 15 parameters of an overparameterized model is an effective strategy”. What this paper does is to show that in a *very specific* setting this is the case, and making a general blanket statement like this is not correct.

Editorial comments:

L 39: “prove that that”

L 64: “MI can be seen as a special case of differential privacy (DP)”
This sentence is not precise. MI is an attack and DP is a defense.

L 90: f_S in the loss should be just f, and the minimization is done with respect to a class of functions (as otherwise there is usually a trivial function to reach loss zero).

L 110: “training algorithm f_S,” -> training algorithm T

Proposition 3.1, just note that the probabilities here are not real probabilities but rather density functions (in case of continuous distributions).

L143: I_d -> L_D

Theorem 3.2 please don’t be too concise: explain the range of i, explain what happens to m=0, and please use parentheses as in (y|m=0,x_0)~N…
right now it took me a while to even get to read the math here due to lack of parentheses.
Also, my calculation gives a different alpha (e.g., the fraction in the logarithm does not have the squared exponents). Please either give a computation or a citation.

Line 160: you cannot simply reverse the variances, you need to flip the decision of the adversary too.

Prop 4.1: why does the last expectation (about the length of x_0,p) exist in this statement? Why not substitute it with the amount of this expectation which can be computed since you pick things from Gaussian.
Also, why does not this proposition appear in Section 4.1? It seems not relevant to Section 4.2 and more relevant to 4.1 .
Also, how is it the case that this proposition “follows immediately as a consequence of Theorem 3.2.”

L 233: “However, as has been shown in a variety of works” I don’t get the relevance of this sentence.

L 252: y should be \hat{y}_0

Many of the graphs have insufficient explanation. For example, for Figure 4, how are n,p,D exactly chosen? Only gamma is given .

Page 8 gets too dense. I find many of the sentences in the last paragraph before Section 5 imprecise and unhelpful. For example “The bottom left in the plot represents the nexus of perfect generalization and no membership advantage”. Where is the bottom left of the plot exactly? Similarly “lower and further to the left in the figure”

Page 9 is also too dense. What is the exact setting of the Time-Series Regression? In particular, is it still the case that an adversary who guesses at random has advantage 0?

L 328: what does “max is taken elementwise” mean?

---

> ### Author Response · Authors · 2022-08-02
> **Response to Reviewer ggEs**
>
> We thank the reviewer for the very detailed feedback on our paper. The reviewer has provided us with numerous insightful comments that we now respond to.
>
> **Regarding methods designed for resisting membership inference (MI)**
>
> We agree with the reviewer that our paper does not guarantee that our findings will hold for methods that employ additional privacy-preserving techniques such as DP-linear regression. Verifying this would be an interesting extension of our current work. Our work is instead focused on studying the effect parameterization has on MI foundationally, and so our analysis centers on a fundamental data model and machine learning algorithm. How this analysis is affected by modern privacy-preserving techniques, especially those commonly used, would certainly be valuable as well, but is out of this paper’s scope. Our paper provides valuable analytical tools that can be used in these future studies.
>
> **On the limitation of the adversary**
>
> Currently, we consider a black-box adversary with one query. The reviewer comments that it would also be valuable to analyze stronger adversaries such as white-box adversaries with multiple queries. We agree that understanding the effect of parameterization for different types of MI adversaries is also interesting. However, the single-query black-box adversary in itself is also a common framework for MI analysis in the literature and is of great interest to the community ([10], [16], [17], [28]). Our paper specifically studies the optimal single-query black-box adversary (likelihood ratio test), which is especially informative in this setting.
>
> Further, many multiple-query attacks use their multiple queries to learn the target model’s output distribution. Since we consider the optimal adversary with full knowledge of this distribution, our adversary does not require multiple queries to obtain this same information.
>
> We do acknowledge that papers also exist analyzing white-box and multiple-query attacks [18] and that there is also much interest in such adversaries. Ultimately, we feel that the choice of adversary is a design choice with both black-box and white-box adversaries being of high interest to the community. We have opted to directly analyze the black-box one for our paper, but we also believe that our analytical tools can be extended towards other types of adversaries in future studies. To consider a white-box adversary, for example, one could modify the distributions of $\hat{y_0}$ in Theorem 3.2 to additionally be conditioned on the parameters of the model. To consider multiple queries, instead of using the distribution of a single value $\hat{y_0}$, one would consider the multidimensional distribution of the vector of query results. We believe our paper provides valuable tools for studying the case of other types of adversaries.
>
> **Regarding over-interpreting our findings**
>
> We fully agree with the reviewer that we do not intend to imply that reducing the number of parameters will universally improve privacy for all data and machine learning models. We will clarify this in our revision and emphasize that our findings are only for the settings we consider. Indeed, it would require more verification to see how these findings hold for other types of models and machine learning algorithms. We do believe that providing theoretical insights for our specific settings still takes a step towards a comprehensive understanding of the general effect of parameterization on membership inference.

---

> > ### Comment · Reviewer_ggEs · 2022-08-08
> > **Ack**
> >
> > Thanks for taking time to answer all of my comments. They were very helpful.
> >
> > Assuming the authors would correct the interpretation of their result (not to claim that more parameters always means more vulnerability) I would stay supportive of this paper and think that it could be a good paper for NeurIPS.

---

> ### Author Response · Authors · 2022-08-02
> **Response to Reviewer ggEs Editorial Comments**
>
> Thank you for the detailed editorial comments!
>
> Thank you for finding typographical errors in our writing. We will fix L39, L110, L143, L252.
>
> Regarding L90, L160:
> Thank you for pointing these out. We will make the necessary clarifications in the revision.
>
> Regarding L64:
> We will change this sentence to the following:
> “Differential privacy (DP) is another popular framework frequently used to study the privacy properties of machine learning algorithms, and models that have DP also enjoy membership inference guarantees [13].”
>
> Regarding Theorem 3.2.
> Thank you for these remarks. We will clarify the statement of this theorem according to these suggestions. As for alpha, the squares in the logarithm come from the ½ factor in the Gaussian distribution function’s exponential term. Our calculation is below, where we begin by setting the two Gaussian distribution functions equal to each other:
> \begin{align*}
>     \frac{1}{\sigma_0 \sqrt{2\pi}}\exp\left({-\frac{1}{2}\left(\frac{\alpha}{\sigma_0}\right)^2}\right) &= \frac{1}{\sigma_1 \sqrt{2\pi}}\exp\left(-\frac{1}{2}\left(\frac{\alpha}{\sigma_1}\right)^2\right) \\\\
>     \frac{\sigma_1}{\sigma_0} &= \exp\left(-\frac{1}{2}\left(\left(\frac{\alpha}{\sigma_1}\right)^2 - \left(\frac{\alpha}{\sigma_0}\right)^2 \right)\right) \\\\
>     \frac{\sigma_1}{\sigma_0} &= \exp\left(-\frac{\alpha^2}{2} \frac{\sigma_0^2 - \sigma_1^2}{\sigma_0^2\sigma_1^2}\right) \\\\
>     \log\left(\frac{\sigma_1}{\sigma_0}\right) &= -\frac{\alpha^2}{2} \frac{\sigma_0^2 - \sigma_1^2}{\sigma_0^2\sigma_1^2} \\\\
>     \alpha &= \sqrt{\frac{2\sigma_0^2\sigma_1^2\log\left(\frac{\sigma_1}{\sigma_0}\right)}{\sigma_1^2 - \sigma_0^2}} \\\\
>     \alpha &= \sqrt{\frac{\sigma_0^2\sigma_1^2\log\left(\frac{\sigma_1^2}{\sigma_0^2}\right)}{\sigma_1^2 - \sigma_0^2}}.
> \end{align*}
>
> Regarding L233:
> Thus far in our paper, we have shown that increasing the number of features decreases the privacy for our model. The sentence in L233 then brings up the contrasting fact that increasing the number of features also improves generalization error, a fact proven and discussed in the references we cited. This leads to a privacy-utility trade-off discussed in the subsequent sentences.
>
> Regarding L328:
> This is a typographical error. We mean to write $\max(\mathbf{Z}\mathbf{V}, 0)$.
>
> Regarding Prop 4.1:
> You are correct that $\mathbb{E}[||x_{0,p}||^2]$ should be replaced with the actual value. We will correct this. This proposition appears in Section 4.2 since it is for the unregularized case studied in 4.2 as opposed to ridge regularization in Section 4.1. This result follows from Theorem 3.2 by noting that the estimate is unbiased and Theorem 3.2 gives the variance in the case that x_0 is drawn freshly from the test set. From there, decomposing mean squared error in terms of bias and variance leads to the statement. We will add clarification on this in our revision.
>
> Regarding Figure 4:
> The values of the variables $(n, p, D)$ are provided in Section 5, and we will mention this in the figure caption.
>
> Regarding pages 8–9:
> Thank you for the feedback. If our paper is accepted, then we will use the additional page to provide additional explanation for these sections. The time-series experiment follows the same general setup in the beginning of Section 5 along with the specifications listed in lines 311–320. All our experiments use Definition 2.1 to measure advantage, for which a random adversary, who would have True Positive Rate = False Positive Rate, would obtain 0 advantage. This includes the time-series regression experiment.

---

### Author Response · Authors · 2022-08-02
**Summary of Paper Contributions**

Thank you to all reviewers for taking the time to review this work and for offering suggestions for improvement. Below, we summarize the key ideas and contributions of this paper to provide some context in our replies to each reviewer.

1. We study the tradeoff between optimal black-box membership inference (MI) attacks and model size.

2. Theoretically, we provide the first rates for MI in overparameterized linear models as a function of the number of features. These rates characterize how quickly vulnerability to MI increases with the number of features.

3. We prove that, surprisingly, ridge regression does not effectively guard against MI.

4. As a means of defending against MI, we study the effectiveness of noise addition as compared to reducing the dimensionality of the model and show that using a smaller model achieves an improved privacy-utility tradeoff vs. noise addition for our data model.

5. Empirically, we demonstrate the model size vs. MI vulnerability phenomenon for various data models (latent-space, time-series, random ReLU features), validating our theoretical results.

---

### Meta-Review · Area_Chair_huTb · 2022-08-23

**Recommendation:** Accept
**Confidence:** Less certain

**Metareview:**

This paper considers the problem of membership inference. The authors propose to use the tools of random matrix theory in the asymptotic regime to analyze membership inference in the simple case of a linear model on Gaussian data. They start by deriving the explicit advantage of the attack in the asymptotic regime. Further derivations allow the author to analyze several interesting machine learning ingredients, starting from L2 regularization (ridge regression). There, the authors show that regularization has the counterintuitive effect of increasing the performance of membership inference attacks. The referees are leaning toward acceptance and I concur.

**Award:**

No

---

### Decision · Program_Chairs · 2022-09-14

Accept